# Central processing of leg proprioception in *Drosophila*

**Sweta Agrawal[1], Evyn S Dickinson[1], Anne Sustar[1], Pralaksha Gurung[1], David Shepherd[2], James W Truman[3,4], John C Tuthill[1]\***

[1]Department of Physiology and Biophysics, University of Washington, Seattle, United States; [2]School of Natural Sciences, Bangor University, Bangor, United Kingdom; [3]Janelia Research Campus, Howard Hughes Medical Institute, Ashburn, United States; [4]Friday Harbor Laboratories, University of Washington, Friday Harbor, United States

**Abstract** Proprioception, the sense of self-movement and position, is mediated by mechanosensory neurons that detect diverse features of body kinematics. Although proprioceptive feedback is crucial for accurate motor control, little is known about how downstream circuits transform limb sensory information to guide motor output. Here we investigate neural circuits in *Drosophila* that process proprioceptive information from the fly leg. We identify three cell types from distinct developmental lineages that are positioned to receive input from proprioceptor subtypes encoding tibia position, movement, and vibration. 13Bα neurons encode femur-tibia joint angle and mediate postural changes in tibia position. 9Aα neurons also drive changes in leg posture, but encode a combination of directional movement, high frequency vibration, and joint angle. Activating 10Bα neurons, which encode tibia vibration at specific joint angles, elicits pausing in walking flies. Altogether, our results reveal that central circuits integrate information across proprioceptor subtypes to construct complex sensorimotor representations that mediate diverse behaviors, including reflexive control of limb posture and detection of leg vibration.

**\*For correspondence:**
tuthill@uw.edu

**Competing interests:** The authors declare that no competing interests exist.

## Introduction

Mechanosensory neurons provide feedback essential for maintaining stable locomotion through unpredictable environments. A subset of these neurons, the proprioceptors, create an internal representation of body state by monitoring joint angles, joint stresses and strains, and muscle length and tension (*Proske and Gandevia, 2012*). Sensory feedback from proprioceptors contributes to many behaviors, including regulation of body posture (*Hasan and Stuart, 1988*; *Zill et al., 2004*), coordination of goal-directed movement (*Büschges, 2005*; *Lam and Pearson, 2002*), locomotor adaptation (*Bidaye et al., 2018*; *Dickinson et al., 2000*), and motor learning (*Isakov et al., 2016*; *Takeoka and Arber, 2019*).

In both invertebrates and vertebrates, proprioceptors encode diverse features of body kinematics (*Tuthill and Azim, 2018*). For example, muscle spindles, which are proprioceptive sensory organs embedded in vertebrate skeletal muscles, encode both muscle fiber length and contraction velocity (*Hunt, 1990*). A functionally analogous structure, the femoral chordotonal organ (FeCO), is housed within the femur of the insect leg (*Field and Matheson, 1998*; *Figure 1A*). The FeCO is the largest proprioceptive organ in the fruit fly, *Drosophila melanogaster* (*Meigen, 1830*), and its 152 neurons can be divided into at least three anatomically distinct subtypes: the claw, hook, and club neurons (*Kuan et al., 2020*; *Mamiya et al., 2018*). Each subtype encodes different kinematic features of the femur-tibia joint: claw neurons encode tibia position (flexion or extension; *Figure 1C*), hook neurons encode directional tibia movement (flexion or extension; *Figure 1D*), and club neurons encode tibia vibration and bidirectional movement (*Figure 1E*). Experimental manipulation of the FeCO in several

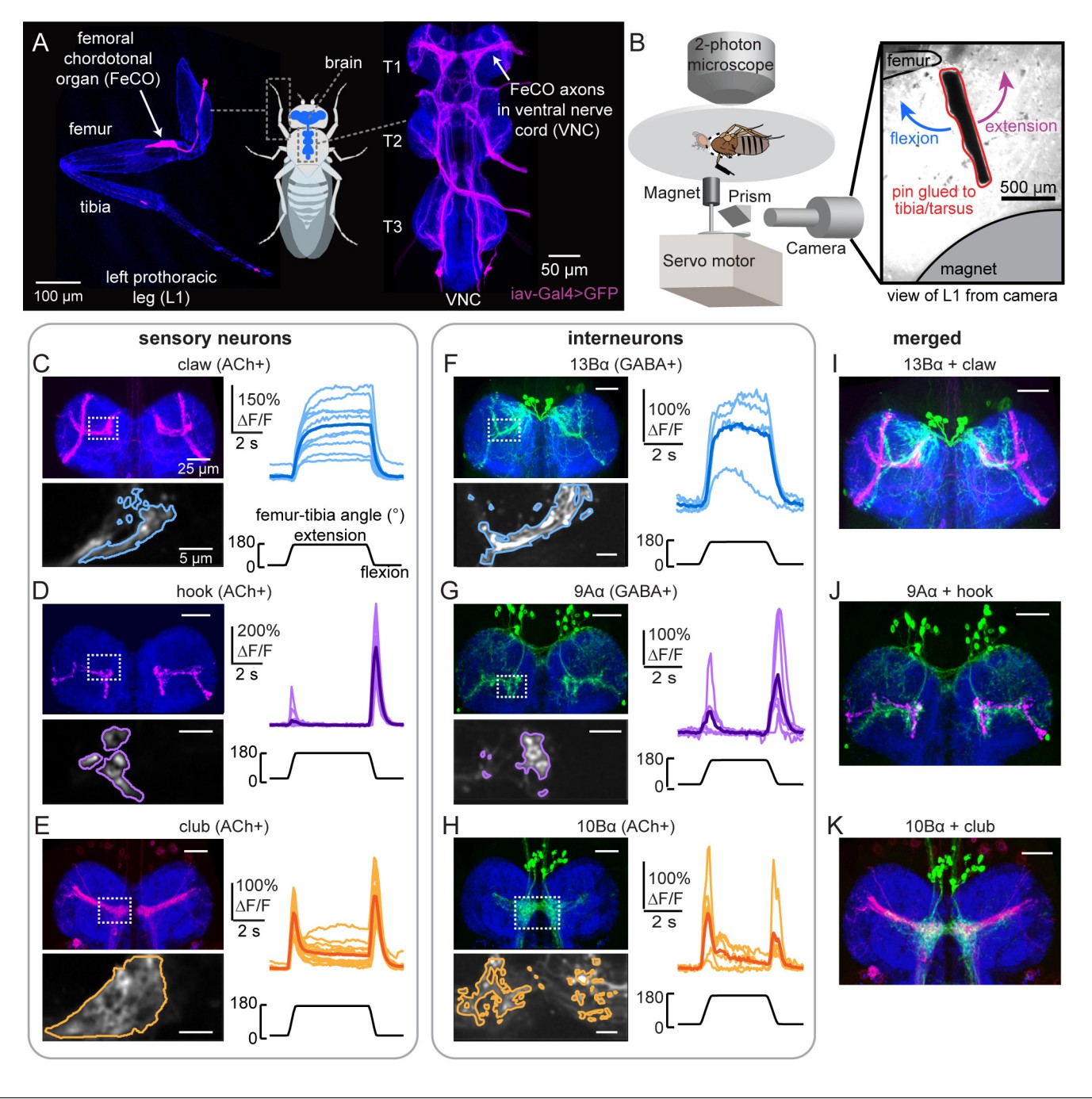

**Figure 1.** Transformation of leg proprioceptive signals from sensory to central neurons. (**A**) Left: Confocal image of the prothoracic (front) leg showing the location of the femoral chordotonal organ (FeCO) cell bodies and dendrites (magenta). Blue: cuticle auto-fluorescence. Right: confocal image of FeCO neurons in the fly ventral nerve cord (VNC). Blue: neuropil stain (nc82); Magenta: FeCO axons. (**B**) Experimental setup for two-photon calcium imaging from VNC neurons while controlling and tracking the femur-tibia joint. A steel pin was glued to the tibia, painted black, and moved via a magnet mounted on a servo motor. The tibia was vibrated by a piezoelectric crystal fixed to the magnet. Right: an example frame from a video used to track joint angle. (**C–H**) Calcium signals from FeCO sensory neurons or central neurons in response to swing movements of the femur-tibia joint. Top left: anatomy (magenta or green) of each cell type in the prothoracic VNC (blue: nc82). The dashed white box indicates the recording region. Bottom left: GCaMP6f fluorescence within the recording region during an example trial. The pixels comprising each region of interest are outlined. Right: changes in GCaMP6f fluorescence (ΔF/F) during femur-tibia swing movements. The thicker line is the response average (n=10, 13, 14, 4, 6, 6). (**I–K**) Overlay of sensory axons (magenta) and central neurons (green). Data in **C–E** were reproduced with permission from *Mamiya et al., 2018*. All VNC images were aligned using the Computational Morphometry Toolkit (*Jefferis et al., 2007*).

*Figure 1 continued on next page*

*Figure 1 continued*

The online version of this article includes the following video and figure supplement(s) for figure 1:

**Figure supplement 1.** Transformation of leg proprioceptive signals by 13Bα, 9Aα, and 10Bα cells in the fly VNC.

**Figure 1—video 1.** *In vivo* calcium imaging from central neurons while manipulating the femur-tibia joint.

https://elifesciences.org/articles/60299#fig1video1

insect species has revealed its importance during behaviors like walking and targeted reaching (*Bässler, 1988*; *Field and Burrows, 1982*; *Mendes et al., 2013*; *Page and Matheson, 2009*).

In contrast to the sensory neurons, nothing is known about how leg proprioceptive signals are combined or transformed by downstream circuits in the adult *Drosophila* central nervous system. Work in other species has shown that proprioceptors synapse directly onto both motor neurons and complex networks of central neurons in the vertebrate spinal cord or invertebrate ventral nerve cord (VNC; *Bässler, 1993*; *Proske and Gandevia, 2012*). Central circuits integrate proprioceptive information across different modalities, muscles, or limbs, and in some cases, integrate proprioceptive information with descending motor commands (*Jankowska, 1992*; *Windhorst, 2007*; *Burrows, 1996*). Ultimately, understanding the role of sensory feedback in motor control will require knowledge about how central neurons transform inputs from limb proprioceptors, as well as their subsequent effect on motor circuits.

Studying the sense of proprioception presents two major challenges. First, proprioception is multimodal: proprioceptors found at the same location in the body can detect different mechanical features produced by self-movement, such as muscle velocity, muscle tension, or joint position (*Proske and Gandevia, 2012*). It is unclear to what degree signals from these diverse proprioceptors are combined to form a composite representation of the body, or whether they are even encoded within a common coordinate system. Additionally, proprioception faces strict constraints on processing speed: in nimble-footed animals like flies, central circuits may have less than 30 ms to process proprioceptive information in between successive steps (*DeAngelis et al., 2019*). Perhaps as a result, proprioceptive and motor circuits are heavily intermingled: many primary and second-order sensory neurons are also premotor neurons that synapse onto motor neurons (*Arber, 2012*; *Büschges and Gruhn, 2007*). The lack of clear hierarchical structure within the spinal cord and VNC has made it challenging to identify general organizational principles of central proprioceptive processing.

To better understand how central circuits process proprioceptive information, we examined how sensory signals from the fly FeCO are transformed by downstream neurons in the VNC. We first used an anatomical screen to identify three neuronal cell types positioned to receive input from at least one of the major FeCO subtypes. We then characterized how each cell type encodes femur-tibia joint kinematics by recording their activity during controlled leg manipulations. Finally, to understand the role of these neurons in motor control, we optogenetically activated each cell type while tracking fly behavior. Our results reveal that, even at this early stage of sensory processing, information from different FeCO subtypes is combined to form diverse, complex representations of tibia movement and position that underlie a range of behaviors, including postural reflexes and vibration sensing.

## Results

The *Drosophila* VNC consists of ~20,000 neurons (*Bates et al., 2019*) that arise from 30 segmentally repeated neuroblasts, each of which divides to form an 'A' and 'B' hemilineage (*Truman et al., 2010*). Developmental lineages are an effective means to classify neuronal cell types: neurons within a hemilineage are morphologically similar (*Harris et al., 2015*; *Mark et al., 2019*), express the same transcription factors (*Allen et al., 2020*; *Lacin and Truman, 2016*), and release the same primary neurotransmitter (*Lacin et al., 2019*). Despite these common features, each hemilineage may be composed of many cell types (*Harris et al., 2015*; *Lacin et al., 2020*), and it remains an open question to what extent neurons within a hemilineage exhibit similar connectivity or function.

We screened a panel of hemilineage-specific split-Gal4 lines for VNC neurons whose dendrites overlap with the axons of FeCO proprioceptors (*Figure 1A*). We computationally aligned VNCs with

GFP expression in sensory and central neurons to assess putative connectivity. Based on this analysis, we focused our efforts on three driver lines that label specific central cell types: (1) a population of GABAergic neurons from the 13B hemilineage (13Bα neurons; *Figure 1F* and *Figure 1—figure supplement 1A*) that are positioned to receive input from position-tuned claw proprioceptors, (2) a population of GABAergic neurons from the 9A hemilineage (9Aα neurons; *Figure 1G* and *Figure 1—figure supplement 1B*) that are positioned to receive input from directionally tuned hook proprioceptors, and (3) a population of cholinergic neurons from the 10B hemilineage (10Bα neurons; *Figure 1H* and *Figure 1—figure supplement 1C*) that are positioned to receive input from vibration-sensitive club proprioceptors. These three cell types are not the only central neurons whose dendrites overlap with FeCO axons – however, they were the top three candidates based on light-level anatomy.

The anatomy of each cell type suggests that they receive input from specific leg proprioceptor subtypes. To test this, we expressed the genetically encoded calcium indicator GCaMP6f in each central neuron population. We then recorded calcium activity *in vivo* via two-photon calcium imaging while using a magnetic control system to manipulate the femur-tibia joint (schematized in *Figure 1B*, from *Mamiya et al., 2018*). We applied three classes of mechanical stimuli to the tibia: swing (*Figure 1C–H* and *Figure 1—figure supplement 1D,G, and J*), ramp-and-hold (*Figure 1—figure supplement 1E,H, and K*), and vibration (*Figure 1—figure supplement 1F,I, and L*).

The calcium responses of each cell type supported our hypothesis that the different populations of VNC neurons process signals from distinct subtypes of FeCO sensory neurons (*Figure 1—video 1*). Similar to extension-tuned claw neurons, 13Bα neurons tonically increased their calcium activity during tibia extension (*Figure 1C and F* and *Figure 1—figure supplement 1D and E*) and were not sensitive to tibia vibration (*Figure 1—figure supplement 1F*). Similar to flexion-tuned hook neurons, 9Aα neurons increased their calcium activity during tibial flexion and to a lesser degree during tibial extension (*Figure 1D and G* and *Figure 1—figure supplement 1G and H*). However, unlike flexion-tuned hook neurons, 9Aα neurons also exhibited large increases in calcium activity during high frequency tibia vibration (*Figure 1—figure supplement 1I*), suggesting that they also integrate signals from vibration-sensitive club neurons. Similar to club neurons, 10Bα neurons transiently increased their activity during tibia extension, flexion, and vibration (*Figure 1E and H* and *Figure 1—figure supplement 1J–L*). Overall, while we lack direct evidence that FeCO sensory neurons provide monosynaptic input to these central neurons, the anatomical proximity and tuning of each cell type are consistent with the hypothesis that they encode tibial movement via input from the FeCO.

## 13Bα neurons linearly encode tibia position via tonic changes in membrane potential

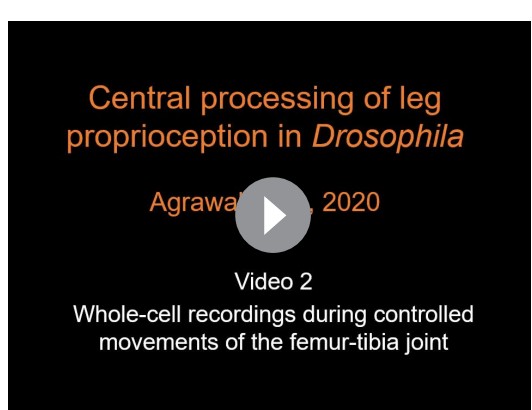

**Video 1.** Whole-cell recordings during controlled movements of the femur-tibia joint. The video shows example whole-cell recordings during controlled movements of the femur-tibia joint for 13Bα, 9Aα, and 10Bα cells.
https://elifesciences.org/articles/60299#video1

Each of the three cell types is comprised of multiple neurons per VNC segment. To assess the heterogeneity of encoding within a cell type, we recorded the activity of single neurons using *in vivo* whole-cell patch clamp electrophysiology (*Figure 2A*). Whole-cell recordings also enabled us to resolve faster time-scale dynamics and determine the contribution of inhibitory inputs, providing insights into the transformations that occur between sensory and central neurons.

Whole-cell recordings from individual 13Bα neurons revealed little heterogeneity across cells. All 13Bα cells lacked detectable action potentials (*Figure 2D*). As suggested by the population-level calcium imaging, the membrane potential of individual 13Bα cells provides a continuous readout of tibial position: each cell depolarizes when the tibia is extended and hyperpolarizes when the tibia is flexed (*Figure 2E and F*, *Video 1*). Aside from a small transient depolarization following joint extension

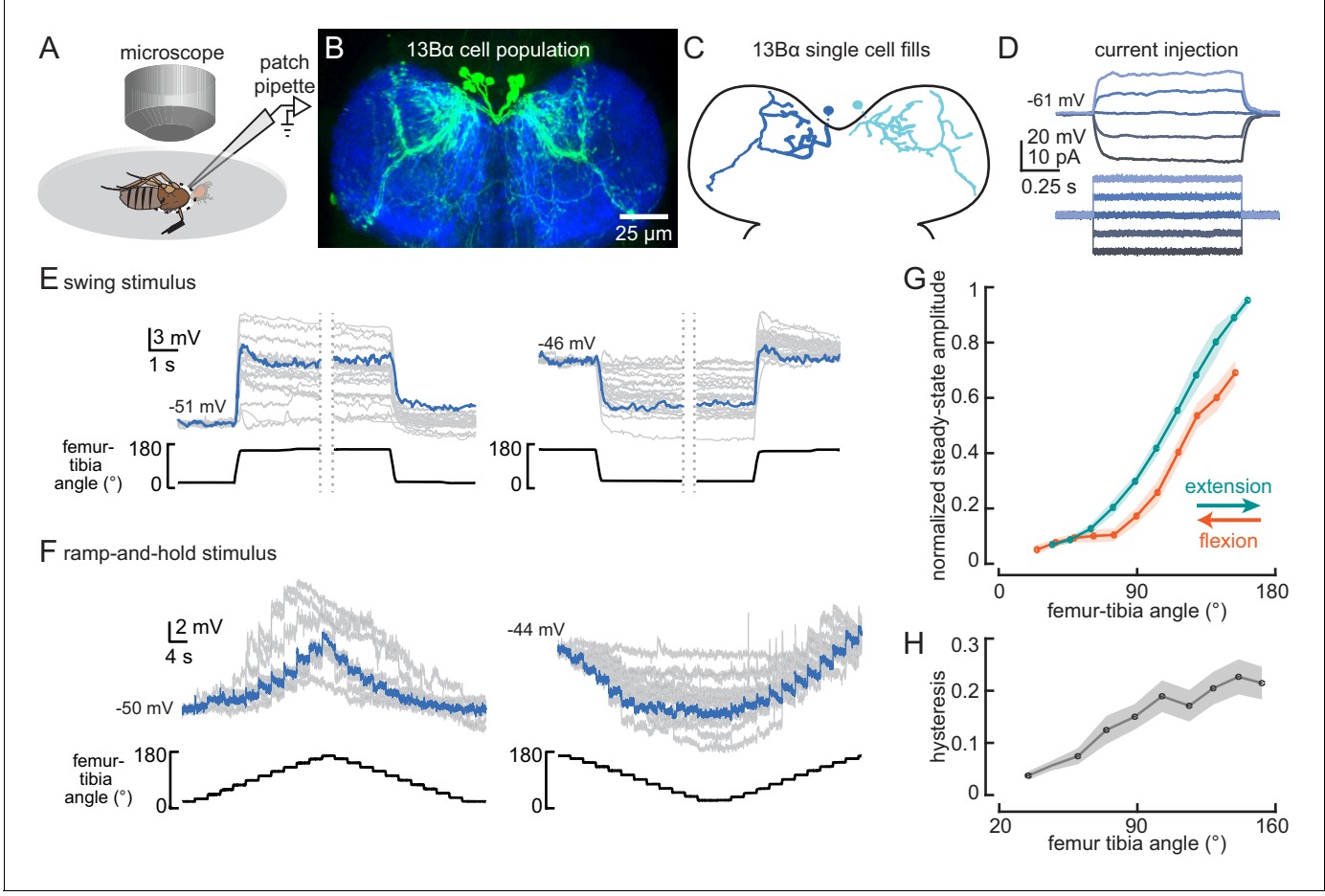

**Figure 2.** 13Bα neurons encode tibia position via tonic changes in membrane potential. (**A**) The experimental setup from *Figure 1B* was modified for whole-cell electrophysiology. (**B**) Confocal image of 13Bα neurons (green) in the prothoracic ventral nerve cord. (**C**) Morphology of two 13Bα neurons reconstructed after filling with Neurobiotin. (**D**) Voltage responses to current injection from an example 13Bα recording. (**E**) Whole-cell current clamp recordings during the indicated swings of the femur-tibia joint. Each trace is the average response of a cell to three presentations of the same movement. An example trace is highlighted in blue (left: n = 19; right: n = 23). (**F**) Current clamp recordings during ramp-and-hold movements of the femur-tibia joint. (left: n = 10; right: n = 15) (**G**) Steady-state activity (average ± SEM) at different joint angles during flexion (orange) or extension (green) measured from ramp-and-hold trials (tibia began fully flexed). Steady-state responses were measured during the middle second of each 3 s step. Individual traces were normalized to the same maximum amplitude. (**H**) Hysteresis (difference between the response to flexion and extension (**G**), average ± SEM) of the steady-state response plotted in **G**.

The online version of this article includes the following figure supplement(s) for figure 2:

**Figure supplement 1.** Pharmacological manipulation of synaptic inputs to 13Bα neurons.

(*Figure 2E*), the responses of 13Bα to movement of the femur-tibia joint were remarkably tonic (i.e., non-adapting) at steady state. These properties were stereotyped across all cells, though there was some cell-to-cell variability in the magnitude of membrane potential fluctuations, perhaps due to variability in recording quality.

The response tuning of single 13Bα neurons was similar to that observed with population-level calcium imaging of extension-tuned claw sensory neurons (*Mamiya et al., 2018*). 13Bα activity increased only when the tibia was extended past ~90˚ (*Figure 2F and G*), and the steady-state membrane potential at a given tibia position was greater when the tibia was extended to reach that position than when the tibia was flexed (*Figure 2G*). This phenomenon, commonly referred to as hysteresis, could introduce ambiguity for downstream neurons that rely on a representation of absolute leg angle. The degree of hysteresis that we observed (*Figure 2H*) is comparable to what has been previously reported for claw neurons (*Mamiya et al., 2018*).

We used pharmacological manipulations to investigate the nature of presynaptic inputs to 13Bα neurons. Bath application of tetrodotoxin (TTX) to the VNC prevents action potential propagation in leg mechanosensory neurons (*Tuthill and Wilson, 2016*). As expected, TTX abolished activity in 13Bα neurons during tibia movement (*Figure 2—figure supplement 1A*). FeCO sensory neurons release the neurotransmitter acetylcholine (*Mamiya et al., 2018*), and so we would expect that application of acetylcholine receptor antagonists (methyllycaconitine [MLA], an antagonist of nicotinic receptors, or atropine, an antagonist of muscarinic receptors) would also block 13Bα activity. Surprisingly, both MLA and atropine had only subtle effects on 13Bα encoding and never completely abolished 13Bα activity (*Figure 2—figure supplement 1B and C*). This result suggests that either 13Bα neurons are coupled to claw sensory neurons via gap junctions, or that MLA and atropine only partially block cholinergic synaptic input from claw neurons.

When the femur-tibia joint moves from an extended to a flexed position, the membrane potential of 13Bα neurons hyperpolarizes to a new steady state. This hyperpolarization could be due to an increase in inhibitory input, a decrease in excitatory input, or a combination of both. Application of picrotoxin, an antagonist of the inhibitory neurotransmitter receptors, $GABA_a$ and $Glu_{Cl}$ (*Liu and Wilson, 2013*; *Wilson and Laurent, 2005*), had no effect on 13Bα activity (*Figure 2—figure supplement 1D*), suggesting that 13Bα neurons do not receive inhibitory input via $GABA_a$ or $Glu_{Cl}$ receptors. Inhibition of 13Bα neurons may instead occur through $GABA_b$ receptors that are not blocked by picrotoxin, though $GABA_b$-mediated conductances are slower (*Wilson and Laurent, 2005*) and therefore unlikely to be involved in encoding rapid joint-angle changes. Further evidence that the hyperpolarization of 13Bα neurons is not mediated by an inhibitory input comes from experiments measuring responses to tibia movement after injecting current to shift the cell's resting membrane potential (*Figure 2—figure supplement 1E and F*). Depolarizing the cell by injecting positive current through the patch pipette shifts the membrane potential toward the equilibrium potential for excitation, reducing the driving force for excitatory synaptic input while increasing the driving force for inhibitory synaptic input. Hyperpolarizing the cell by injecting negative current has the opposite effect. We found that when 13Bα cells were depolarized, the change in membrane potential during both extension and flexion decreased (*Figure 2—figure supplement 1E*), suggesting that the proprioceptive responses of 13Bα neurons are mediated by increases and decreases in excitatory input. In summary, 13Bα cells are a relatively homogeneous class of neurons that receive excitatory input from extension-sensitive claw neurons, perhaps via mixed chemical and electrical synapses.

## Activation of 13Bα neurons causes tibia flexion

Our measurements of 13Bα activity demonstrate that these neurons encode extension of the femur-tibia joint. Their tonic, non-adapting responses suggest a role in encoding, and potentially controlling, posture of the femur-tibia joint. To test this hypothesis, we optogenetically activated 13Bα neurons in tethered, headless flies. We expressed the light-gated cation channel CsChrimson (*Klapoetke et al., 2014*) in 13Bα neurons and used a green laser focused on the ventral thorax at the base of the left front (L1) leg to activate neurons in the left prothoracic VNC (*Figure 3A and B*). Because we were interested in whether these neurons drive reflexive leg movements, we measured how optogenetic activation altered movements of the three major leg joints (coxa-femur, femur-tibia, and tibia-tarsus) in headless flies with their legs unloaded (i.e. the fly was suspended in the air; *Figure 3A*) or loaded (i.e. the fly was positioned on a ball; *Figure 3B*). In the absence of descending signals from the brain, decapitated flies maintain a consistent leg posture but rarely move their legs spontaneously (*Figure 3—figure supplement 1A and B*). In both loaded and unloaded flies, activation of 13Bα neurons caused a slow extension of the coxa-femur joint and flexion of the femur-tibia joint; this movement was absent during trials without a laser stimulus (*Figure 3C and D* and *Figure 3—video 1*). Prior experiments using the same behavioral setup demonstrated that a laser stimulus in the absence of CsChrimson does not cause leg movement in headless flies (*Azevedo et al., 2020*). During some trials we also observed a lateral movement of the middle left (L2) leg (*Figure 3—figure supplement 1A and B* and *Figure 3—video 1*). The movement of both joints was larger in unloaded flies, whereas the lateral movement of L2 was more likely to occur in loaded flies. For those flies that flexed their femur-tibia joint, the change in joint angle (*Figure 3E and F* and *Figure 3—figure supplement 1C and D*) and likelihood of flexion (*Figure 3G and H*) did not vary with initial joint position. Thus, despite systematic differences in initial joint positions of loaded and

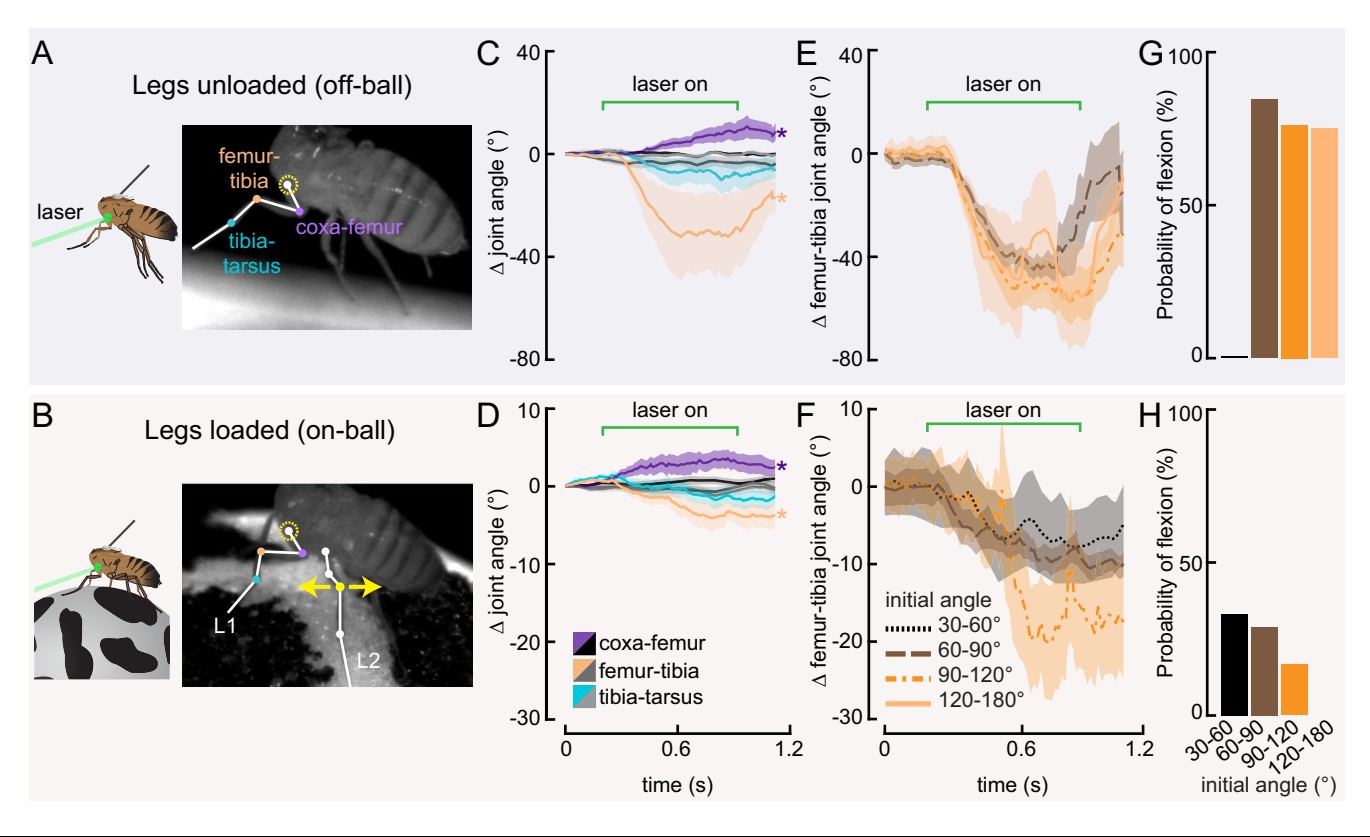

**Figure 3.** Optogenetic activation of 13Bα neurons causes flexion of the femur-tibia joint. (**A and B**) Schematic (left) and example frame (right) illustrating optogenetic activation of 13Bα neurons in headless flies either suspended from a tether (legs unloaded, **A**) or positioned on a ball (legs loaded, **B**). A green laser (530 nm) is focused at the coxa-body joint of the fly's left front leg (outlined in yellow) and all leg joints are monitored with high-speed video. Yellow arrows illustrate the left middle leg's lateral movements. Top row of panels: legs unloaded; Bottom row: legs loaded. (**C and D**) Average change in joint angle (± SEM) of the coxa-femur (purple/black), femur-tibia (orange/dark gray), or tibia-tarsus (blue/light gray). Colored traces are from trials with a 720 ms laser stimulus (as indicated by the green bracket), and the black and gray traces are from trials with no laser stimulus. Asterisks mark those leg joints that demonstrated a significant change in joint angle when the laser was on compared to the no-laser trials (*p<0.05, bootstrapping with false discovery rate correction). Fly is either unloaded (**C**, n = 4 flies) or loaded (**D**, n = 7 flies). (**E and F**) Average change in the femur-tibia joint angle during laser stimulation grouped by initial joint angle. Only trials in which the fly flexed the tibia are included. Fly is either unloaded (**E**, n = 30 trials) or loaded (**F**, n = 19 trials). None of the groups was significantly different from one another (p<0.05, one-way ANOVA with Tukey–Kramer correction for comparisons across multiple populations). (**G-H**) Probability that we observed a femur-tibia flexion across all trials. Bars are color coded according to initial joint angle as in **E** and **F**. G: n = 43 trials, F: n = 74 trials.

The online version of this article includes the following video and figure supplement(s) for figure 3:

**Figure supplement 1.** Optogenetic activation of 13Bα neurons causes movement of the L1 and L2 legs.

**Figure 3—video 1.** Optogenetic activation of 13Bα neurons in headless flies.

https://elifesciences.org/articles/60299#fig3video1

unloaded flies (*Figure 3—figure supplement 1E and F*), we hypothesize that the distinct responses to optogenetic stimulation were due to the activity of other proprioceptors, such as campaniform sensilla, that are activated by leg loading (*Zill et al., 2004*). Overall, our results suggest that 13Bα neurons mediate slow postural leg movements in response to limb perturbations detected by the FeCO. Such leg movements are similar to resistance reflexes caused by manipulation of the FeCO in other insects (*Field and Matheson, 1998*).

## 9Aα cells exhibit cell-to-cell diversity in their encoding of tibial flexion

The anatomy of 9Aα neurons suggests that they receive input from the directionally tuned hook neurons (*Figure 4A and B*). Whole-cell recordings confirmed this hypothesis, but also revealed unexpected levels of heterogeneity in the 9Aα population. Each 9Aα cell we recorded from responded

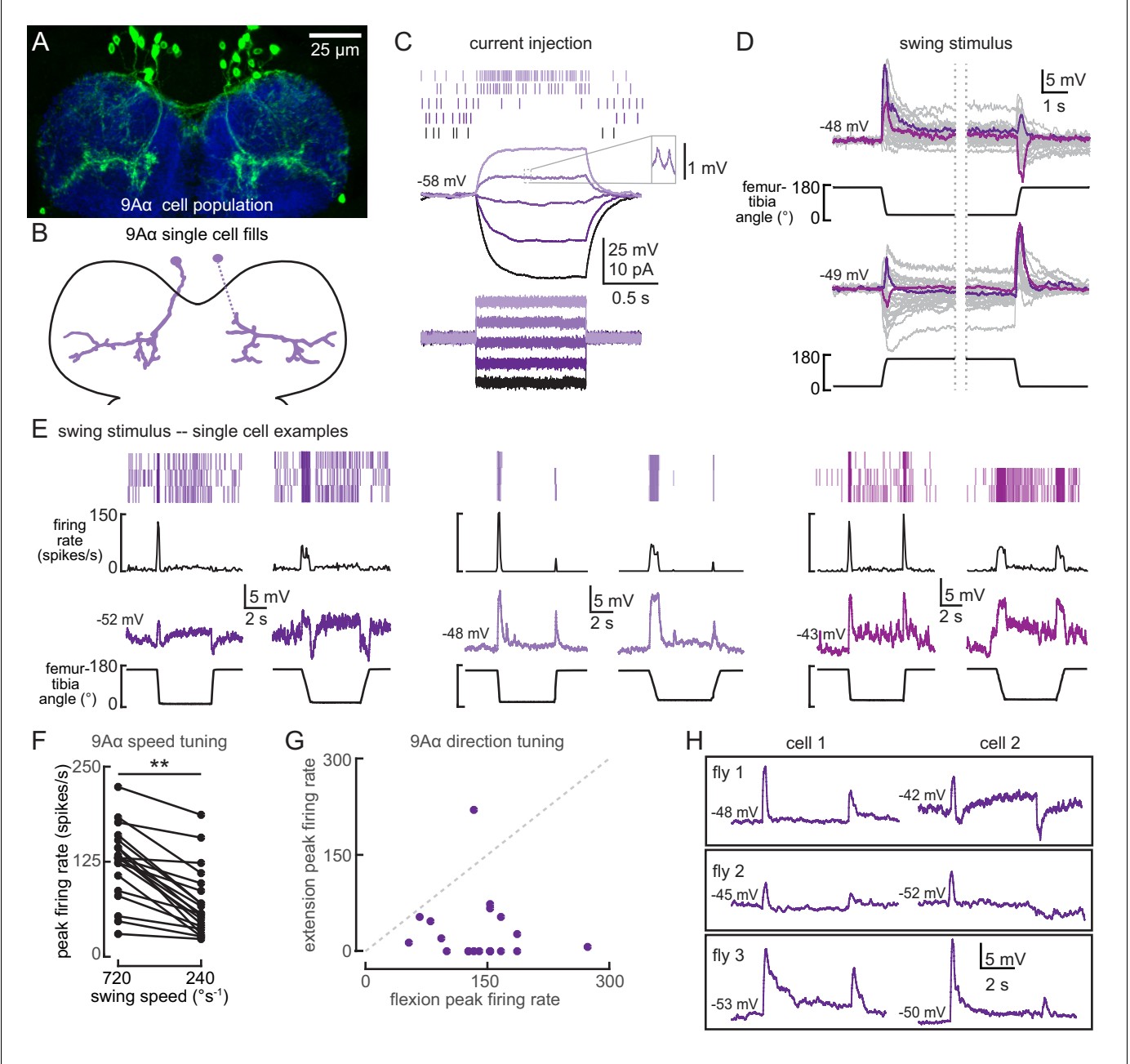

**Figure 4.** 9Aα neurons are a functionally heterogenous population that encode tibia movement direction. (**A**) Confocal image of 9Aα neurons (green) in the prothoracic ventral nerve cord (VNC). (**B**) Morphology of two 9Aα neurons reconstructed after filling with Neurobiotin. (**C**) Voltage responses to current injection from an example 9Aα recording. Detected spikes are indicated above the voltage traces. Inset shows example spikes, enlarged for clarity. (**D**) Whole-cell current clamp recordings during tibia swing. Each trace is the averaged response to three stimulus presentations. Two example traces are highlighted in purple and magenta (top: n = 35, bottom: n = 27). (**E**) Example whole-cell current clamp recordings from three cells during fast ($720°s^{-1}$) and slow ($240°s^{-1}$) swings. Each pair of traces is recorded from a single cell. (**F**) Peak firing rates (averaged across three stimulus presentations) for different flexion speeds (**p<0.005, Wilcoxon matched-pairs signed-rank test). (**G**) For each cell, the peak firing rate when the tibia was flexed relative to when the tibia was extended. Points along the dashed line would represent cells that are equally sensitive to both directions of movement. Points found below the dashed line are tuned for flexion. (**H**) Pairs of 9Aα cells in the same fly have distinct responses to the same $720°s^{-1}$ swing movement.

The online version of this article includes the following figure supplement(s) for figure 4:

**Figure supplement 1.** 9Aα cells are a heterogenous population of neurons that encode tibial flexion.

to tibia movement through changes in membrane potential and action potential firing rate (*Figure 4C and D*). Although individual neurons had consistent tuning across the duration of a recording, each 9Aα cell had slightly different response tuning. The only consistent properties of 9Aα neurons were their directional and speed tuning: subthreshold and spiking activity were largest during fast, flexing swing movements (*Figure 4E–G Figure 4—figure supplement 1*, and *Video 1*). Other properties were more variable. For example, some cells were inhibited by tibial extension (*Figure 4E*, left), other cells were excited by tibial extension (*Figure 4E*, middle and right), and some cells were also tonically depolarized when the tibia was held flexed (*Figure 4E*, right, S3B). This latter observation suggests that some 9Aα cells receive direct or indirect inputs from position-sensitive claw neurons. We confirmed that the diversity in 9Aα encoding was not simply due to fly-to-fly variability by recording from two 9Aα neurons in the same fly (*Figure 4H*).

Pharmacology experiments suggest that 9Aα cells receive both inhibitory and excitatory inputs. Similar to 13Bα cells, TTX blocked 9Aα encoding of leg movement (*Figure 4—figure supplement 1D*). However, unlike 13Bα cells, MLA also blocked proprioceptive responses in 9Aα neurons (*Figure 4—figure supplement 1E*), indicating that 9Aα activity requires acetylcholine release from FeCO sensory neurons. In several neurons that were hyperpolarized by tibial extension, picrotoxin application abolished the hyperpolarization of membrane potential, suggesting that GABAergic inhibition contributes to the encoding of tibial extension (*Figure 4—figure supplement 1F*).

Whole-cell recordings confirmed that 9Aα cells also respond to high frequency tibia vibration (*Figure 5*). Because hook neurons are not sensitive to tibia vibration (*Mamiya et al., 2018*), this observation suggests that 9Aα cells receive direct or indirect input from vibration-sensitive club neurons. Again, we found cell-to-cell heterogeneity in 9Aα vibration encoding. Some cells were inhibited by lower frequency vibration, and as a result, more sharply frequency-tuned (*Figure 5A*). Cells also varied in their rates of adaptation: some exhibited a sustained vibration response (*Figure 5B*) whereas others adapted quickly after vibration onset (*Figure 5A*). Nevertheless, every 9Aα cell was maximally depolarized by 1600–2000 Hz vibrations, and response magnitude increased at higher vibration amplitudes (*Figure 5C*). During a small number of recordings, MLA application abolished the vibration response (*Figure 5D*), suggesting that 9Aα vibration encoding also requires acetylcholine release from FeCO sensory neurons. Picrotoxin application abolished inhibitory responses to lower frequency vibration (*Figure 5E*). Picrotoxin application during these few recordings also reduced the amplitude of 9Aα responses to high frequency vibration. This decrease may be caused by a reduced excitatory conductance after picrotoxin depolarized the resting membrane potential. Thus, in addition to direction-tuned inputs from FeCO hook neurons, 9Aα cells also receive vibration-sensitive inputs from FeCO club neurons.

VNC hemilineages contain multiple cell types with diverse projection patterns (*Harris et al., 2015*). Most of our recordings from 9A neurons targeted just one 9A cell type, the 9Aα cells. However, twice when recording from the driver line labeling 9Aα neurons, we recorded from a cell that was morphologically and physiologically distinct. This cell, which we refer to as 9Aα2, has a cell body located within the same cluster as other 9A neurons, but its neurites extend anteriorly, similar to claw axons (*Figure 5—figure supplement 1C*). 9Aα2 cells have larger spikes than 9Aα cells (>2 mV), and they encode flexed tibial positions via tonic changes in membrane potential and firing rate (*Figure 5—figure supplement 1A and B*). 9Aα2 cells did not respond to tibial vibration (*Figure 5—figure supplement 1D*), and MLA application mostly blocked their responses to tibia movement (*Figure 5—figure supplement 1E*). This result suggests that the 9A hemilineage broadly integrates sensory input from the FeCO, and different cell types within the 9A hemilineage receive input from different FeCO sensory neurons.

## Activation of 9Aα neurons causes extension of the tibia-tarsus and femur-tibia joints

Our recordings revealed that 9Aα neurons encode tibia flexion and high-frequency vibration. We next tested if, like 13Bα neurons, optogenetic activation of 9Aα neurons in the left prothoracic VNC (*Figure 6A and B*) would cause leg movements in headless flies. Optogenetic activation of 9Aα neurons produced small extensions of the tibia-tarsus and femur-tibia joints in headless flies standing on a ball (legs loaded; *Figure 6C and D* and *Figure 6—video 1*). We tested a second split-Gal4 line that also labels 9Aα cells (9Aα-L2-Gal4; *Figure 6—figure supplement 1*), and observed similar extensions of the femur-tibia and tibia-tarsus joints (*Figure 6—figure supplement 1B and C*). Both

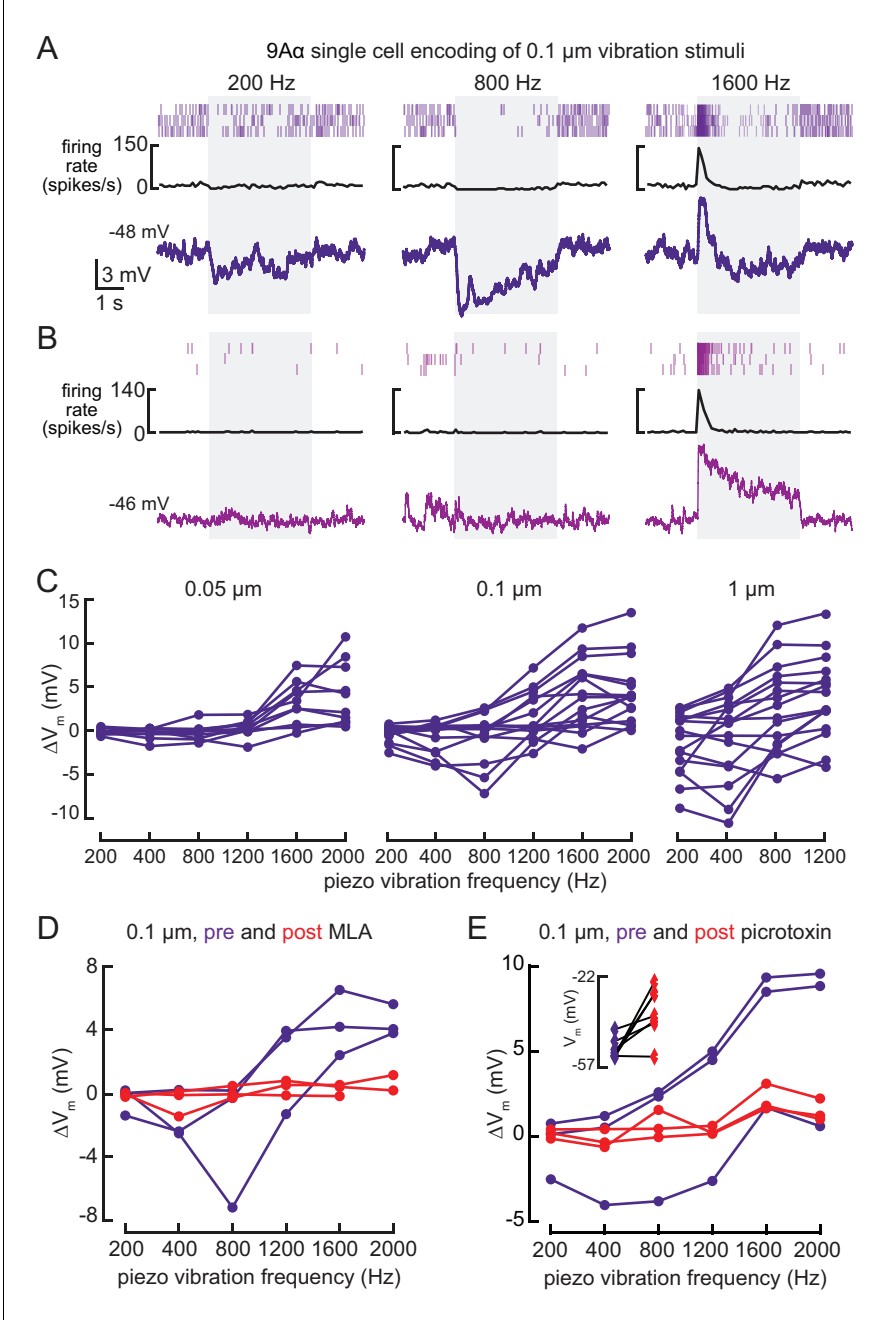

**Figure 5.** 9Aα neurons encode high frequency tibia vibration. (**A and B**) Example whole-cell current clamp recordings from two 9Aα cells during a 0.1 µm tibia vibration. The shaded region indicates the duration of the vibration stimulus. (**C**) The change in membrane potential during the first 500 ms of vibration across amplitudes and frequencies. Each point is the averaged response of a cell to three stimulus presentations (from left to right, n = 10, 15, 16). (**D and E**) The change in membrane potential during the first 500 ms after vibration onset before (purple) and after (red) application of an antagonist of nicotinic acetylcholine receptors, MLA (1 µM, **D**), or before (purple) and after (red) application of the GABA a and GluCl antagonist, picrotoxin (100 µM, **E**). The inset in E shows the resting membrane potential before (purple) and after (red) application of picrotoxin. Picrotoxin application depolarized the membrane potential in the majority of cells (n = 7).

The online version of this article includes the following figure supplement(s) for figure 5:

**Figure supplement 1.** 9Aα2 neurons encode flexed tibia positions.

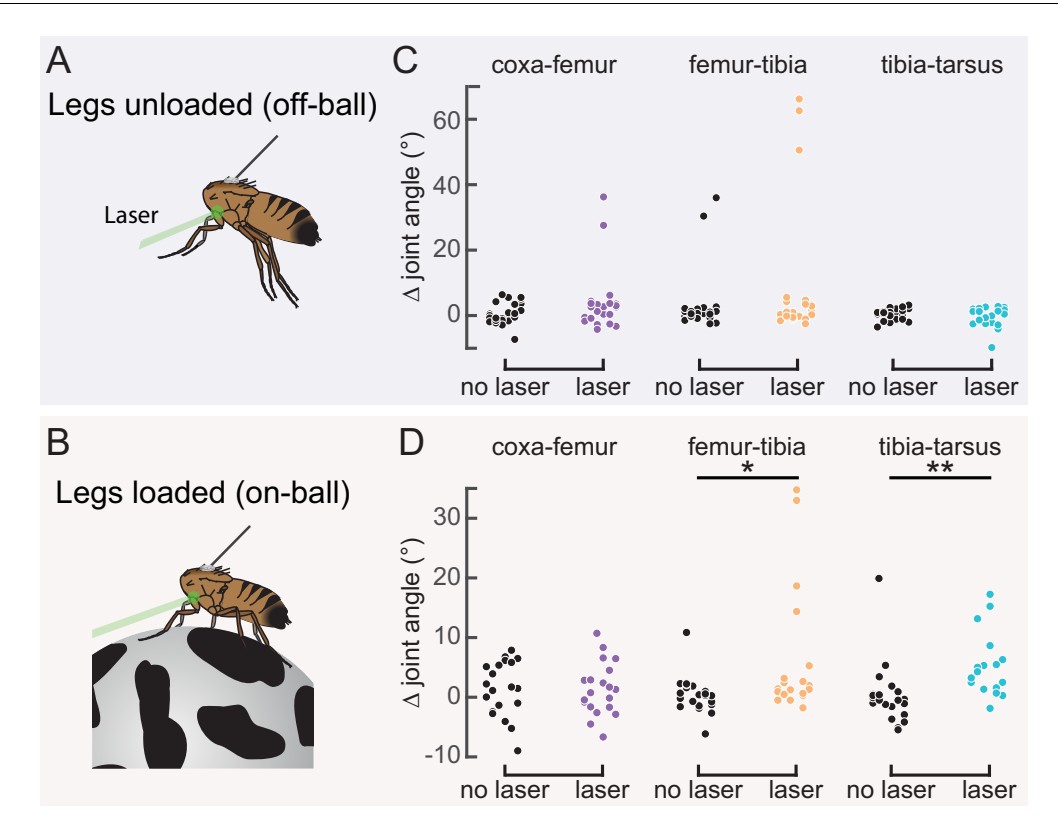

**Figure 6.** Optogenetic activation of 9Aα neurons causes extension of the tibia-tarsus. (**A and B**) Schematic of optogenetic activation of 9Aα neurons (using 9Aα-Gal4) in headless flies either suspended from a tether (legs unloaded, **A**) or positioned on a spherical treadmill (legs loaded, **B**). A green laser (530 nm) is focused at the coxa-body joint of the fly's left front leg. (**C and D**) Change in joint angle after 720 ms during trials in which the laser was on (purple, orange, and blue) or off (black). Each column is data from a single fly. Left: coxa-femur joint; Middle: femur-tibia joint; Right: tibia-tarsus joint. The fly was either unloaded (**C**, n = 6 flies) or loaded (**D**, n = 6 flies). *p<0.05, **p<0.005, bootstrapping with false discovery rate correction.

The online version of this article includes the following video and figure supplement(s) for figure 6:

**Figure supplement 1.** Optogenetic activation of 9Aα neurons using another split-Gal4 line causes a similar extension of the femur-tibia and tibia-tarsus.

**Figure 6—video 1.** Optogenetic activation of 9Aα neurons in headless flies.

https://elifesciences.org/articles/60299#fig6video1

driver lines may also label one or more 9Aα2 cells, meaning we may be activating other 9A cell types. However, calcium imaging, electrophysiology, and GFP expression all suggest that 9Aα are the predominant cells labeled by both driver lines. As with 13Bα neurons, the differences we saw between loaded and unloaded flies could be due to the activity of other proprioceptors activated by leg loading, or because of systematic differences in initial leg posture (*Figure 6—figure supplement 1D and E*). Compared to the 13Bα neurons, the leg movements caused by 9Aα activation were smaller and more variable. Thus, while both neural populations likely mediate postural adjustments in response to limb perturbations detected by the FeCO, 9Aα neurons may do so in a context-dependent manner, for example to produce small corrective movements during walking.

## 10Bα neurons integrate information about tibia vibration and position

10Bα neurons are anatomically positioned to receive input from the axons of FeCO club sensory neurons in the VNC (*Figure 7A and B*). Each 10Bα neuron innervates multiple VNC segments, and a subset of 10Bα cells project up into the central brain, where they innervate the antennal motor and mechanosensory center (AMMC). We used whole-cell patch-clamp electrophysiology to record the

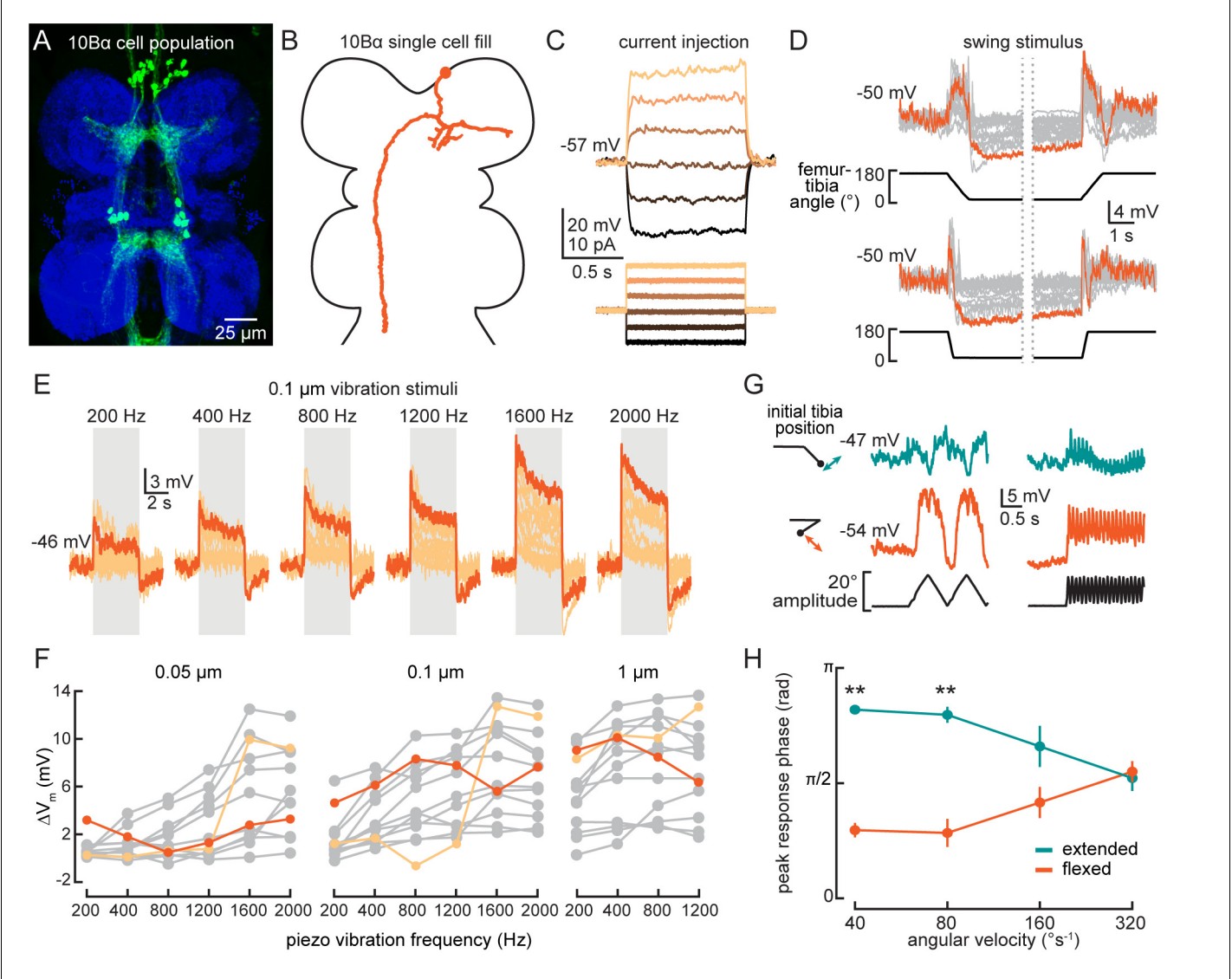

**Figure 7.** 10Bα neurons encode bidirectional tibia motion and tibia vibration in a position-dependent manner. (A) Confocal image of 10Bα neurons (green) in the ventral nerve cord (VNC). (B) 10Bα morphology reconstructed after filling with Neurobiotin. (C) Voltage responses to current injection from an example recording. (D) Whole-cell current clamp recordings during tibia swing movements. Each trace is the average response to three stimulus repetitions. An example trace is highlighted in orange (top: n = 18, bottom: n = 18). (E) Responses to a 0.1 μm vibration stimulus (n = 12 cells). The gray box indicates when the vibration stimulus was applied. An example response is highlighted in dark orange. (F) The change in membrane potential during the first 500 ms after vibration onset. Each point is the averaged response to three stimulus repetitions. Two cells with different frequency tuning are highlighted in different shades of orange (left to right, n = 9, 13, 13). (G) Example 10Bα recording demonstrating how the responses of a single 10Bα neuron to the same movement stimulus depend on the tibia's position. The tibia began either extended (green) or flexed (orange) and was then oscillated with an amplitude of 20° at four different angular velocities. Responses from the slowest (40˚s⁻¹) and fastest (320˚s⁻¹) oscillation are shown. The phase of the oscillation at which a 10Bα cell is maximally depolarized when the tibia began either flexed (orange) or extended (green) (n = 9, **p<0.005, Wilcoxon matched-pairs signed-rank test).

The online version of this article includes the following figure supplement(s) for figure 7:

**Figure supplement 1.** 10Bα cells encode bidirectional tibia movements and tibia vibration.

membrane potential of 10Bα neurons with cell bodies in the T1 segment. In our recordings, current injection failed to evoke identifiable action potentials (*Figure 7C*), though we did occasionally observe spike-like events. Because these events only occurred in a subset of recordings, we instead analyzed changes in the membrane potential of 10Bα neurons during tibial movements.

Consistent with our hypothesis that 10Bα neurons are downstream of FeCO club neurons, individual cells were transiently depolarized by tibia movements in both directions (*Figure 7D*, *Figure 7—figure supplement 1B–C*, and *Video 1*). Additionally, most 10Bα neurons were tonically hyperpolarized when the tibia was fully flexed and transiently hyperpolarized when the tibia was fully extended (*Figure 7D*). Thus, in addition to movement-sensitive excitatory inputs, 10Bα cells also receive position-tuned inhibition.

Like club sensory neurons, we found that 10Bα neurons are sensitive to low amplitude and high frequency vibration of the tibia. Interestingly, different 10Bα cells were tuned to different ranges of vibration frequency (*Figure 7E and F*). As vibration amplitude increased, frequency tuning broadened (*Figure 7F*, light orange) or shifted (*Figure 7F*, dark orange). 10Bα cells are so sensitive that they responded to vibration caused by the saline perfusion system (*Figure 7—figure supplement 1A*, left inset). This perfusion response was absent when the tibia was flexed (*Figure 7—figure supplement 1A*, right inset), suggesting that position-dependent inhibition of 10Bα neurons is sufficient to suppress vibration encoding. Thus, leg position may modulate flies' ability to sense substrate vibration via the FeCO.

Tibia position also modulated the sensitivity and timing of 10Bα activity during larger amplitude movements. When we applied an identical 20° triangle-wave oscillation to the tibia starting at either an extended (~145°) or flexed (~20°) position, tibia position affected both the amplitude and phase of the resulting membrane potential oscillations (*Figure 7G and H*). This phase shift decreased as the oscillation frequency increased, and disappeared during movements faster than 320°s$^{-1}$. As a result, the effect of tibia position on the timing of 10Bα activity may only be significant during slower movements like grooming or targeted reaching. During faster movements like walking, tibia position would primarily modulate the amplitude of 10Bα activity, not its timing.

Finally, pharmacology experiments suggest that 10Bα neurons, like 13Bα neurons, may be electrically coupled to upstream FeCO sensory neurons. Application of acetylcholine antagonists (MLA or atropine) was not sufficient to disrupt 10Bα encoding of tibia swing (*Figure 7—figure supplement 1E and F*) or tibia vibration (*Figure 7—figure supplement 1H*). Application of picrotoxin abolished the tonic hyperpolarization present during tibia flexion for some cells (*Figure 7—figure supplement 1G*), suggesting that 10Bα neurons receive inhibitory inputs. Application of picrotoxin also decreased responses to vibration and abolished the vibration offset response (*Figure 7—figure supplement 1I*). In summary, 10Bα neurons are intersegmentally projecting central neurons that encode tibia movement and vibration via input from club sensory neurons. Their vibration sensitivity is gated by inhibition that depends on the position of the tibia.

## 10Bα neurons drive pausing behavior in walking flies

10Bα neurons are sensitive to leg movements detected by the FeCO, but optogenetically activating 10Bα neurons in headless flies did not reliably evoke leg movement (data not shown). This result suggests that, unlike 9Aα and 13Bα neurons, 10Bα neurons do not modulate leg postural adjustments.

Behavioral studies of walking flies demonstrate that vibration of the substrate can cause flies to stop walking (*Fabre et al., 2012*; *Howard et al., 2019*). To determine if 10Bα neurons could drive this pausing behavior, we optogenetically activated 10Bα neurons in tethered, intact flies walking on a spherical treadmill (*Figure 8A*). As with headless flies, we used a green laser focused on the base of the left front leg to activate neurons in the left prothoracic VNC. As an optogenetic control, we used flies that expressed only the Gal4 activation domain but not the DNA-binding domain (split-half(SH)-Gal4). These flies have a similar genetic background as the split-Gal4 lines labeling the VNC interneurons, but they lack expression of a functional Gal4 protein or CsChrimson. Comparing control flies and CsChrimson-expressing flies allowed us to distinguish behavioral responses that were due to a reaction to the laser (which is within the spectral range of the fly's vision) from those due to optogenetic activation.

Activating 10Bα neurons in walking flies consistently led to flies slowing or stopping after about 200 ms, regardless of the length of the laser stimulus (*Figure 8B–D*, *Figure 8—figure supplement 1A*, and *Figure 8—video 1*). Although control flies also sometimes paused during the stimulus period, flies with activated 10Bα neurons paused earlier and more frequently (*Figure 8D*). Activating 13Bα also caused flies to slow (*Figure 8B* and *Figure 8—figure supplement 1B and C*), likely due to movement of the front leg, which interrupted walking. Despite their vibration sensitivity, activating

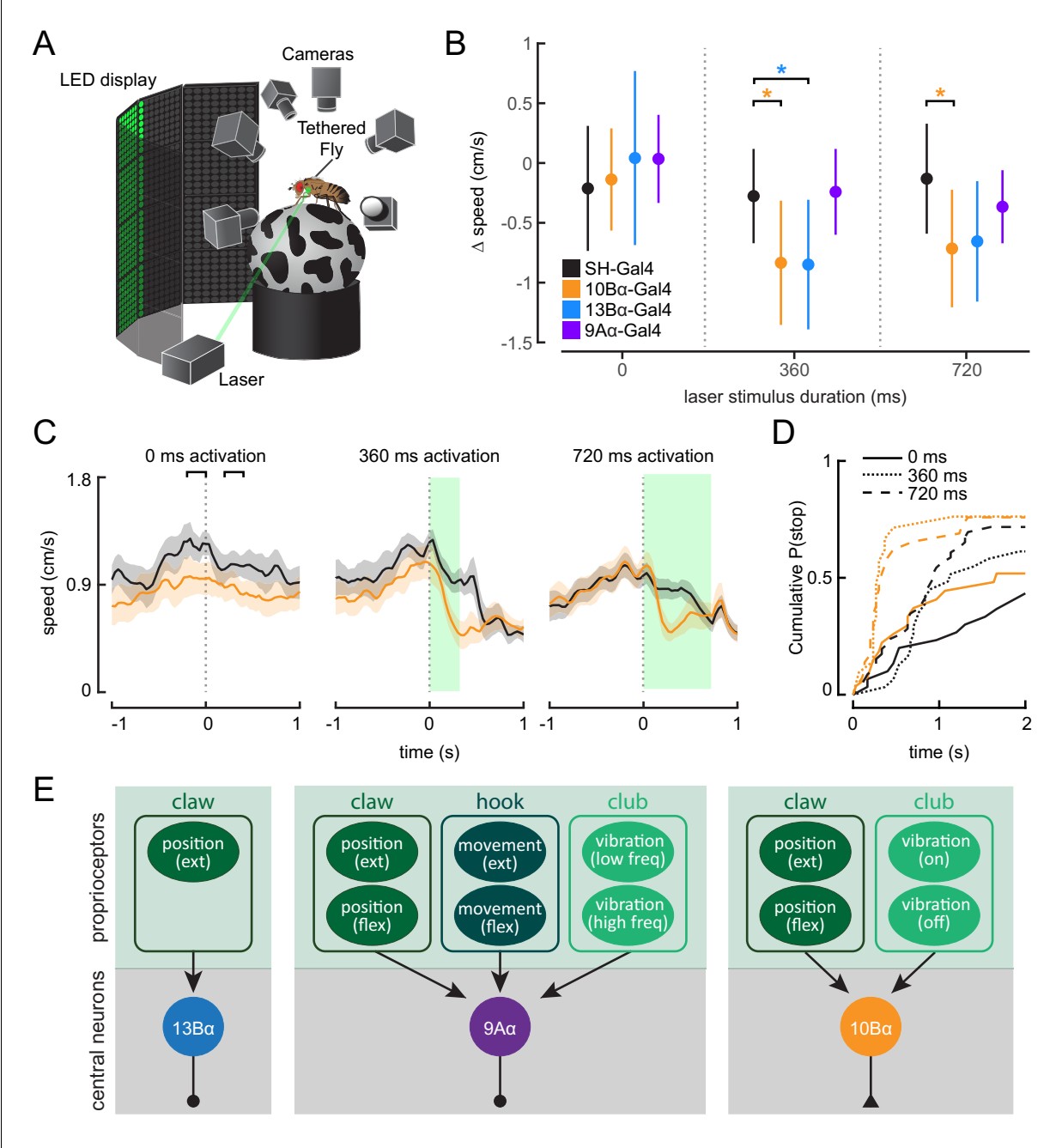

**Figure 8.** Optogenetic activation of 10Bα neurons causes flies to freeze and stop walking. (**A**) Schematic of optogenetic activation of 10Bα neurons in tethered flies walking on a spherical treadmill. The treadmill and fly are tracked using high-speed cameras, and an LED display presents visual patterns that encourage walking. A green laser (530 nm) is focused on the coxa of the fly's left front leg. (**B**) Difference in speed during the 200 ms preceding the start of the stimulus period compared to a 200 ms window beginning after the start of the stimulus period, as indicated by the brackets in **C** (*p<0.05, bootstrapping with false discovery rate correction; 0 ms: Control: n = 6 flies; 10Bα: n = 10 flies; 13Bα: n = 7 flies; 9Aα: n = 7 flies; 360 ms: Control: n = 8 flies; 10Bα: n = 10 flies; 13Bα: n = 8 flies; 9Aα: n = 6 flies; 720 ms: Control: n = 13 flies; 10Bα: n = 11 flies; 13Bα: n = 13 flies; 9Aα: n = 11 flies). (**C**) Average treadmill forward velocity (± SEM) of walking flies during no laser trials (left) or trials with a 360 ms (middle) or 720 ms (right) laser stimulus. Green boxes indicate the duration of optogenetic stimulation. Black brackets indicate the pre- and post-laser onset time periods used to compare the effect of activation between control and interneuron lines in **B**. (**D**) Cumulative probability of a fly stopping (velocity <0.3 cm/s) during trials with no laser or trials with a 360 ms or 720 ms laser stimulus (0 ms: Control: 25 trials, 10Bα: 24 trials; 360 ms: Control: 25 trials, 10B: 19 trials; 720 ms: Control: 42 trials, 10Bα: 24 trials). (**E**) Schematic of the three central neuron populations (13Bα, 9Aα, and 10Bα) and their sensory inputs as determined by our experiments.

The online version of this article includes the following video and figure supplement(s) for figure 8:

*Figure 8 continued on next page*

*Figure 8 continued*

**Figure supplement 1.** Optogenetic activation of 10Bα neurons causes flies to freeze and stop walking.
**Figure 8—video 1.** Optogenetic activation of 10Bα neurons in walking flies.
https://elifesciences.org/articles/60299#fig8video1

9Aα neurons had no effect on flies' walking velocity (*Figure 8B*, and *Figure 8—figure supplement 1B and C*).

Overall, our physiology and behavior data indicate that 10Bα neurons trigger pausing in response to tibia vibration detected by the FeCO. Thus, we propose that vibration-detecting club FeCO neurons and their downstream partners, the 10Bα neurons, may comprise a pathway for sensing external substrate vibration. However, 9Aα neurons, which drive reflexive leg movements but not pausing, also respond to tibia vibration. This result suggests that encoding of tibia vibration by club neurons contributes to both exteroceptive and proprioceptive mechanosensory processing.

## Discussion

Understanding the role of proprioceptive feedback in motor control requires knowledge about how central neurons transform inputs from limb proprioceptors and their subsequent effect on motor circuits. In this study, we found that proprioceptive information from diverse limb proprioceptors is relayed to central neurons in the *Drosophila* VNC that process these signals in parallel (*Figure 8E*). Some neurons, like 13Bα, encode only a single kinematic feature, tibia extension, presumably via input from the extension-sensitive claw neurons, whereas other neurons, like 9Aα and 10Bα neurons encode complex combinations of tibia movement, high frequency vibration, and tibia position, presumably via inputs from multiple proprioceptor subtypes. These central neurons contribute to a range of behaviors, including postural reflexes and vibration sensing, which depend on the animal's behavioral context. Overall, our results suggest that the logic of sensory integration in second-order proprioceptive circuits may be best understood with respect to their motor function.

### Comparison with proprioceptive circuits in other insects

Neurons similar to 9Aα or 13Bα neurons have been identified in other insects (*Burrows, 1988*; *Burrows, 1996*; *Büschges, 1990*). For example, the nonspiking I1 and I2 neurons in the stick insect encode femur-tibia joint position in a manner similar to 13Bα neurons (*Büschges, 1990*) and local spiking neurons in the locust encode a mixture of joint flexion and position similar to 9Aα neurons (*Burrows, 1988*). However, it is difficult to determine whether these populations are homologous based only on their physiology or anatomy. FeCO sensory neurons vary from species to species in their number, mechanosensory sensitivity, and axonal projections (*Collin, 1985*; *Mamiya et al., 2018*; *Matheson, 1992b*). Additionally, due to differences in size, leg shape, walking gait, and ecological niche, central integration of proprioceptive information may be species-specific (*Büschges and Wolf, 1995*; *Field and Matheson, 1998*). Although there exist many compelling similarities between our results and those in larger orthopteran insects, we cannot currently make clear conclusions about cell-type homology. Nevertheless, it will be interesting to see if, as in the stick insect and locust, activity of nonspiking neurons such as 13Bα in *Drosophila* can reset walking phase (*Büschges, 1995*) or adjust the gain of local reflexes (*Laurent and Burrows, 1989*).

### A developmental framework for identifying functional subunits in the insect VNC

In vertebrates, knowledge of developmental lineages has provided a useful framework for understanding the physiology and function of neurons in the spinal cord (*Catela et al., 2015*; *Jessell, 2000*; *Lu et al., 2015*). We have undertaken a similar approach, beginning with previously described lineage maps of the *Drosophila* VNC (*Harris et al., 2015*; *Shepherd et al., 2016*) to identify central neurons that are positioned to receive sensory information from the FeCO. We then built genetic driver lines to label these neurons. Because we still lack quantitative data on the numbers of cells or cell types within each hemilineage, it is unclear what proportion of a given cell type is captured by each driver line. Nevertheless, as has been observed in other species (*Shepherd and*

*Laurent, 1992*; *Thompson and Siegler, 1991*), we found that neurons from the same hemilineage possess similarities in their neurophysiological properties, encoding of tibial kinematics, and putative connections with upstream FeCO neurons. Thus, knowledge of a cell's developmental origins can be a powerful means to identify functional microcircuits within the fly VNC.

*Drosophila* is a holometabolous insect that undergoes metamorphosis, changing from a larva to an adult fly. These two life stages look and behave differently, but their nervous systems are generated by the same segmental array of neuroblasts (*Harris et al., 2015*; *Lacin and Truman, 2016*). Neurons arising from the same neuroblast produce a similar set of molecules and innervate similar nerve tracts in both larvae and adults (*Birkholz et al., 2015*; *Lacin and Truman, 2016*) – are they also functionally similar? Several central neurons in the larval VNC, such as the Basin neurons, have been implicated in relaying mechanosensory or proprioceptive input to motor circuits (*Heckscher et al., 2015*; *Jovanic et al., 2016*; *Mark et al., 2019*; *Ohyama et al., 2015*; *Zarin et al., 2019*). Basin neurons, like adult 9A neurons, descend from lineage nine and receive inputs from larval chordotonal neurons – do adult 9A neurons also receive nociceptive input or synapse onto motor neurons? Such comparisons will yield insight into how central circuits for proprioception are repurposed following metamorphosis.

Beyond understanding how neural function is conserved across metamorphosis, understanding the relationship between hemilineage identity and circuit function will also reveal how neural circuits are conserved across evolution. The organization of neuroblasts that gives rise to the insect VNC has undergone little change over 350 million years of insect evolution (*Lacin and Truman, 2016*; *Thomas et al., 1984*; *Truman and Ball, 1998*) and homologous neuroblasts and their resulting lineages have been identified in insects as diverse as silverfish, grasshoppers, and *Drosophila* (*Jia and Siegler, 2002*; *Thomas et al., 1984*; *Truman and Ball, 1998*; *Witten and Truman, 1998*). Connecting neurons' functions with their developmental origin in different species will yield a powerful framework for studying the evolution of sensorimotor circuits, revealing the essential bauplan underlying flexible, fast motor control.

## Central integration of proprioceptive sensory information

We found that VNC neurons integrate mechanosensory signals from proprioceptor subtypes that encode distinct features of leg joint kinematics. Convergence of multiple kinematic features has also been described in second-order neurons in other mechanosensory systems. For example, aPN3 neurons, a class of neurons downstream of the Johnston's organ (JO) in the fly antenna, encode antennal vibrations only at specific antennal positions, similar to 10Bα neurons (*Chang et al., 2016*; *Patella and Wilson, 2018*). Second-order neurons within the mammalian vestibular nuclei include cells that encode head-rotational movements (via inputs from the semi-circular canal), head-translational movements (via inputs from the otolith), or both rotations and translations (*Dickman and Angelaki, 2002*; *Goldberg, 2000*). The continuum from unimodal to complex, multimodal encoding is thought to facilitate the vestibular nuclei's role in multiple behaviors, including vestibular-ocular reflexes and disambiguating translational motion from gravitational accelerations (*Angelaki and Cullen, 2008*; *Green and Angelaki, 2010*; *Green et al., 2005*). Similarly, the range of second-order neurons that we found in the VNC are likely shaped by the constraints of processing speed and need for motor flexibility.

Each VNC cell type we analyzed had a different degree of functional heterogeneity across individual neurons. 9Aα neurons, in particular, demonstrated high cell-to-cell variability in their response tuning. These diverse response profiles could result from different mixtures of inhibitory and excitatory synaptic inputs, similar to what has been observed in aPN3 neurons in the *Drosophila* antennal mechanosensory circuit (*Chang et al., 2016*). By mixing inhibitory and excitatory inputs in different ratios from different populations of sensory neurons, individual aPN3 neurons demonstrate diverse tuning profiles with sensitivity for different stimulus features. From an information coding perspective, such heterogeneous populations could enable continuous representation of multi-modal stimulus spaces, and encode increased information as a population (*Azarfar et al., 2018*). Functional heterogeneity within a cell type has not been systematically analyzed in second-order proprioceptive neurons of larger insects (*Burrows, 1996*), likely due to the inability to unambiguously assign cell-type identity when recording from unlabeled neurons.

## Neural representation of tibia position

Tibia position is encoded by approximately 25 claw sensory neurons per leg, each of which is tuned to a narrow range of femur-tibia joint angles (*Mamiya et al., 2018*). Claw neurons can be separated into two subtypes encoding either flexed (0–90°) or extended (90–180°) tibia angles. 13Bα neurons, based on their anatomy and activity, are likely downstream of extension-sensitive claw neurons (*Figures 1* and *2*) and the 9Aα2 neurons are a complementary population of inhibitory neurons that receive inputs from flexion-sensitive claw neurons (*Figure 4—figure supplement 1*). Unlike movement-encoding 9Aα or 10Bα neurons, these position-encoding central neurons encode information from only a single FeCO subtype, suggesting that movement information is immediately contextualized by position information, whereas position information can be transmitted independently. Perhaps as a result, optogenetic activation of 13Bα neurons consistently caused leg movements, whereas the effect of 9Aα activation was more variable.

We also found that 13Bα and claw neurons exhibit a similar degree of hysteresis: the steady-state membrane potential at a given tibia position from 90 to 180° was about 20% greater when the tibia was extended to reach that position than when it was flexed (*Figure 2G*). Proprioceptive hysteresis is found in many vertebrate and invertebrate mechanosensory systems (*Grigg and Greenspan, 1977*; *Lennerstrand, 1968*; *Matheson, 1992a*; *Ridgel et al., 2000*). However, it is unclear if hysteresis causes problematic ambiguities for downstream circuits that require an accurate readout of tibia angle, or if it is a useful feature, perhaps compensating for the nonlinear properties of muscle activation in short sensorimotor loops (*Zill and Jepson-Innes, 1988*). Non-spiking central neurons in locusts (*Siegler, 1981a*) and stick insects (*Büschges, 1990*) also exhibit hysteresis, and the effects of hysteresis can be seen in leg motor neuron activity (*Field and Burrows, 1982*; *Siegler, 1981b*). These data suggest that hysteresis is preserved within central circuits, and our results show that the same is true in *Drosophila*.

## Behavioral function of central proprioceptive neurons

Activating both 13Bα and 9Aα neurons caused flies to move their legs, suggesting that these two populations mediate leg postural reflexes in response to perturbations detected by the FeCO. Such reflexes are important to stabilize posture by maintaining joint position, and work in other insects has shown that they are mediated via pathways between the FeCO and leg motor neurons (*Burrows, 1996*; *Büschges, 1990*). Motor neurons controlling the fly tibia are organized according to a hierarchical gradient of cellular size and electrical excitability that enables motor neurons controlling weak, slow movements to be recruited first, followed by neurons that control progressively stronger, faster movements (*Azevedo et al., 2020*). We found that 13Bα activation caused generally larger leg movements than 9Aα activation. While this distinction could be due to differences in how strongly we were able to excite either population, it may also suggest that 13Bα and 9Aα cells provide input to cells at different levels of the motor neuron hierarchy. 13Bα neurons could provide input to higher gain intermediate or fast motor neurons, whereas 9Aα neurons may provide input to only low gain, slow motor neurons.

In contrast to 9Aα and 13Bα neurons, activating 10Bα neurons did not produce reflexive leg movements. Instead, 10Bα neurons drive pauses in walking behavior. Previous behavioral experiments found that flies will stop walking when they sense the ground vibrating (*Fabre et al., 2012*; *Howard et al., 2019*) – 10Bα neurons may mediate this stopping via vibration-sensitive inputs from the FeCO. If true, this would imply that the FeCO functions as both a proprioceptive and exteroceptive organ. This finding is consistent with work in stick insects and locusts that also found that vibration-tuned FeCO neurons do not contribute to postural reflexes (*Field and Pflüger, 1989*; *Kittmann et al., 1996*; *Stein and Sauer, 1999*), and instead mediate startle responses to substrate vibration (*Friedel, 1999*; *Stritih Peljhan and Strauß, 2018*; *Takanashi et al., 2016*). Similar mechanoreceptors that primarily sense substrate vibrations are also found in the limbs of rodents (*Prsa et al., 2019*).

A subset of 10Bα neurons send ascending projections to the brain, where they innervate the wedge (*Figure 1—figure supplement 1C*), a region that encodes auditory information from the antennae (*Patella and Wilson, 2018*). These ascending projections raise the possibility that vibration signals from 10Bα neurons are integrated with vibration signals from the antennae. Although the purpose of this integration is not yet clear, one possibility is that leg vibration could sensitize flies to

other auditory stimuli. Interestingly, however, vibration encoding by the FeCO is not purely extero-ceptive: activating vibration-sensitive 9Aα neurons did not cause walking flies to pause, but did cause flies to extend the tibia-tarsus joint. Thus, vibration coding in the FeCO may be used for both exteroceptive detection of substrate vibration and for proprioceptive feedback control of limb movement.

## Summary

In this study, we identify and describe the physiology, activity, and behavioral function of three populations of central neurons that are positioned to receive synaptic input from proprioceptive sensory neurons in the fly leg. While these cell types represent only a subset of the neurons that are downstream of the FeCO, they provide valuable insights into how proprioceptive sensory information is integrated by central neurons to influence locomotion and motor control. Already, even at the earliest stages of sensory processing, proprioceptive signals from the FeCO diverge to multiple neuron types with distinct behavioral roles.

What is the significance of the specific representations of femur-tibia joint kinematics encoded by these central neurons? Answering this question will ultimately require understanding how the outputs of these neurons feed into motor circuits. *Drosophila* is a uniquely powerful model system for this kind of circuit dissection: recent efforts have identified and mapped the majority of leg motor neurons (*Azevedo et al., 2020*; *Baek and Mann, 2009*; *Brierley et al., 2012*) and leg sensory neurons (*Kuan et al., 2020*; *Mamiya et al., 2018*; *Tsubouchi et al., 2017*; *Tuthill and Wilson, 2016*). Additionally, serial-section electron microscopy of the VNC will enable synapse-resolution reconstruction of the sensorimotor connectome (*Maniates-Selvin et al., 2020*). This solid anatomical framework, coupled with detailed functional investigations of VNC cell types such as the one undertaken in this study, will deepen our understanding of the fundamental computations underlying proprioception.

# Materials and methods

**Key resources table**

| Reagent type (species) or resource | Designation | Source or reference | Identifiers | Additional information |
|---|---|---|---|---|
| Genetic reagent (*D. melanogaster*) | 'w[1118]; P{JFRC7-20XUAS-IVS-mCD8::GFP}attp40' | Other | FBrf0212432 | 'Barret Pfeiffer, Janelia Farm, HHMI' |
| Genetic reagent (*D. melanogaster*) | 'P{iav-Gal4.K}3' | Bloomington *Drosophila* Stock Center | 'RRID:BDSC_52273' | |
| Genetic reagent (*D. melanogaster*) | 'P{GMR73D10-GAL4}attP2' | Bloomington *Drosophila* Stock Center | 'RRID:BDSC_39819' | |
| Genetic reagent (*D. melanogaster*) | 'P{20xUAS-IVS-GCaMP6f} attP40; P{w[+mC]=UAS tdTom.S}3' | Other | N/A | 'Gift from Peter Weir and Michael Dickinson' |
| Genetic reagent (*D. melanogaster*) | 'P{GMR21D12-GAL4}attP2' | Bloomington *Drosophila* Stock Center | 'RRID:BDSC_48946' | |
| Genetic reagent (*D. melanogaster*) | 'P{GMR64C04-GAL4}attP2' | Bloomington *Drosophila* Stock Center | 'RRID:BDSC_39296' | |
| Genetic reagent (*D. melanogaster*) | 'P{28A12-p65.AD} attp40; P{VT000606-GAL4.DBD}attP2' | Other | N/A | 'James Truman and David Shepherd' |
| Genetic reagent (*D. melanogaster*) | 'P{52E12-p65.AD} attp40; P{VT044946-GAL4.DBD}attP2' | Other | N/A | 'James Truman and David Shepherd' |

*Continued on next page*

*Continued*

| Reagent type (species) or resource | Designation | Source or reference | Identifiers | Additional information |
|---|---|---|---|---|
| Genetic reagent (*D. melanogaster*) | 'P{VT043132-p65.AD} attp40; P{VT045623-GAL4.DBD}attP2' | Other | N/A | 'James Truman and David Shepherd' |
| Genetic reagent (*D. melanogaster*) | 'P{25G05-GAL4}attP2' | Bloomington *Drosophila* Stock Center | 'RRID:BDSC_49137' | |
| Genetic reagent (*D. melanogaster*) | 'w[1118]; P{10A07-p65.AD} attp40' | Bloomington *Drosophila* Stock Center | 'RRID:BDSC_69465' | |
| Genetic reagent (*D. melanogaster*) | 'P{y[+t7.7] w[+mC]=20 XUAS-IVS-CsChrimson.mVenus}attP2' | Bloomington *Drosophila* Stock Center | 'RRID:BDSC_55136' | |
| Genetic reagent (*D. melanogaster*) | 'w[1118]; P{VT014013-p65.AD} attp40; P{30A10-GAL4.DBD}attP2' | Other | N/A | 'James Truman and David Shepherd' |
| Antibody | nc82 (mouse monoclonal) | Developmental Studies Hybridoma Bank | 'RRID:AB_2314866' | '(1:50)' |
| Antibody | Anti-CD8 (rat monoclonal) | Thermo Fisher Scientific | 'RRID:AB_10392843' | '(1:50)' |
| Antibody | Goat anti-rat secondary antibody, Alexa Fluor 488 conjugate | Thermo Fisher Scientific | 'RRID:AB_2534074' | '(1:250)' |
| Antibody | Goat anti-mouse secondary antibody, Alexa Fluor 633 conjugate | Invitrogen | 'RRID:AB_141431' | '(1:250)' |
| Chemical compound, drug | MLA | Tocris | TOCRIS_1029 | '(1 μM)' |
| Chemical compound, drug | Atropine sulfate | Sigma-Aldrich | 1045009 | '(20 μM)' |
| Chemical compound, drug | TTX | Abcam | ab120055 | '(1 μM)' |
| Chemical compound, drug | Picrotoxin | Sigma-Aldrich | P1675 | '(100 μM)' |
| Software, algorithm | MATLAB | Mathworks | 'RRID:SCR_001622' | |
| Software, algorithm | FIJI | 'PMID:22743772' | 'RRID:SCR_002285' | |
| Software, algorithm | Fictrac | 'DOI:10.1016/j.jneumeth.2014.01.010' | | |

*Continued on next page*

*Continued*

| Reagent type (species) or resource | Designation | Source or reference | Identifiers | Additional information |
|---|---|---|---|---|
| Software, algorithm | Computational Morphometry Toolkit | Neuroimaging Informatics Tools and Resources Clearinghouse | | 'https://www.nitrc.org/projects/cmtk/' |
| Software, algorithm | ScanImage 5.2 | Vidrio Technologies | 'RRID:SCR_014307' | |
| Other | Streptavidin-Alexa Fluor | Thermo FisherScientific | 'FISHER: S11226' | '(1:250)' |
| Other | Green CST DPSS laser | Besram Technology, Inc | | '(532 nm)' |

## Fly husbandry

*Drosophila* were raised on cornmeal agar food on a 14 hr dark/10 hr light cycle at 25°C. Females flies, 1–3 days post eclosion, were used for all electrophysiology experiments. Female flies, 4–8 days post eclosion, were used for all *in vivo* calcium imaging experiments. For tethered behavior experiments, both male and female flies, between 2 and 10 days post-eclosion, were used. For experiments involving optogenetic reagents (CsChrimson), adult flies were placed on cornmeal agar with all-trans-retinal (35 mM in 95% EtOH, Santa Cruz Biotechnology) for 24 hr prior to the experiment. Vials were wrapped in foil to reduce optogenetic activation during development.

## Fly preparation for *in vivo* two-photon calcium imaging or electrophysiology

To gain optical access to the VNC while moving the tibia, we used one of two previously described fly holders: for calcium imaging experiments, we used the holder as described by *Mamiya et al., 2018*, whereas for electrophysiology experiments, we used the holder as described by *Tuthill and Wilson, 2016*; *Figure 1B*. Flies were anesthetized on ice and then positioned ventral side up, with the head glued to the upper side of the fly holder using UV-cured glue (Bondic or Kemxert 300). We further glued the ventral side of the thorax onto the hole and on the bottom side of the holder, we glued down the femur of the experimental leg (the right prothoracic leg for the majority of experiments, unless otherwise indicated) so that we could control the femur-tibia joint angle by moving the tibia. When gluing the femur, we held it at a position where the movement of the tibia during the rotation of the femur-tibia joint was parallel to the plane of the fly holder. To eliminate mechanical interference, we glued down all other legs. We also pushed the abdomen to the left side and glued it at that position, so that the abdomen did not block tibia flexion. To position the tibia using the magnetic control system described below, we cut a small piece of insect pin (length ~1.0 mm, 0.1 mm diameter; Living Systems Instrumentation) and glued it onto the tibia and the tarsus of the right prothoracic leg. To enhance contrast and improve tracking of the tibia/pin position, we painted the pin with either black India ink (for calcium imaging experiments, Super Black, Speedball Art Products) or white acrylic paint (for electrophysiology experiments, Liquitex heavy body acrylic, titanium white). After immersing the ventral side of the preparation in extracellular fly saline (recipe below), we removed the cuticle above the prothoracic segment of the VNC and took out the digestive tract to reduce the movements of the VNC. We also removed fat bodies and larger trachea to improve access to the leg neuropil. The perineural sheath under the hole was removed for electrophysiological recordings but left intact for calcium imaging. Fly saline contained 103 mM NaCl, 3 mM KCl, 5 mM TES, 8 mM trehalose, 10 mM glucose, 26 mM $NaHCO_3$, 1 mM $NaH_2PO_4$, 1.5 mM $CaCl_2$, and 4 mM $MgCl_2$ (pH 7.1, osmolality adjusted to 270–275 mOsm). Recordings were performed at room temperature.

## Image acquisition using a two-photon excitation microscope

We used a modified version of a custom two-photon microscope previously described in detail (*Euler et al., 2009*). For the excitation source, we used a mode-locked Ti/sapphire laser (Mira 900 F,

Coherent) set at 930 nm and adjusted the laser power using a neutral density filter to keep the power at the back aperture of the objective (40×, 0.8 NA, 2.0 mm wd; Nikon Instruments) below ~25 mW during the experiment. We controlled the galvo laser scanning mirrors and the image acquisition using ScanImage software (version 5.2) within MATLAB (MathWorks). To detect GCaMP6f and tdTomato fluorescence, we used an ET510/80M (Chroma Technology Corporation) emission filter (GCaMP6f) and a 630 AF50/25R (Omega optical) emission filter (tdTomato) and GaAsP photomultiplier tubes (H7422P-40 modified version without cooling; Hamamatsu Photonics). We acquired images (256 × 120 pixels or 128 × 240 pixels) at 8.01 Hz. At the end of the experiment, we acquired a z-stack of the labeled neurons to confirm the recording location.

## Image processing and calculating ΔF/F

We performed all image processing and analyses using scripts written in MATLAB (MathWorks) (https://github.com/sagrawal/InterneuronAnalysis). After acquiring the images for a trial, we first applied a Gaussian filter (size 5 × 5 pixel, s = 3) and aligned each frame to a mean image of the trial using a sub-pixel registration algorithm (*Guizar-Sicairos et al., 2008*) (registered to ¼ pixel). For alignment of images, we used the red channel tdTomato fluorescence, which should not change as a function of calcium. tdTomato fluorescence remained stable over the course of each experiment (data not shown), indicating that movement artifacts were absent or small. For detecting calcium signals, we chose pixels whose mean GCaMP6f fluorescence was above a set threshold (see *Figure 1C–H* for examples). For calculating the GCaMP6f fluorescence change relative to the baseline (ΔF/F), we used the lowest average fluorescence level in a 10-frame window as the baseline fluorescence during that trial.

## CNS electrophysiology

Cell bodies were visualized using an 850 nm IR LED (M850F2, ThorLabs) and a 40× water-immersion objective (Nikon) on an upright fluorescence microscope (SOM, Sutter Instruments). Extracellular saline was bubbled with 95% $O_2$/5% $CO_2$. The internal solution for whole-cell recordings was composed of (in mM) 140 KOH, 140 aspartic acid, 10 HEPES, 2 mM EGTA, 1 KCl, 4 MgATP, 0.5 $Na_3$GTP, 13 Neurobiotin, with pH adjusted using KOH to 7.2 and osmolality adjusted to 268 mOsm. Whole-cell patch pipettes were pulled with a P-97 linear puller (Sutter Instruments) from borosilicate glass (OD 1.5 mm, ID 0.86 mm) to have approximately 8–12 MOhm resistance. Whole-cell patch-clamp recordings were targeted to GFP-labeled cell bodies in the prothoracic region of the VNC. We used a Multiclamp 700A amplifier (Molecular Devices) for all recordings. Data were low-pass filtered at 5 kHz before they were digitized at 20 kHz by a 16 bit A/D converter (Axon Digidata 1400A, Molecular Devices Co) and acquired in AxoScope 10.7 (Molecular Devices). Stable recordings were typically maintained for 1–2 hr. Analysis of electrophysiology data was performed with custom scripts written in MATLAB (MathWorks). The liquid junction potential for the whole cell recordings was −12 mV (*Gouwens and Wilson, 2009*). We corrected the membrane voltages reported in the paper by post-hoc subtraction of the junction potential.

## Moving the tibia/pin using a magnetic control system

We used a previously described magnetic control system (*Mamiya et al., 2018*) to manipulate the femur/tibia joint angle. To move the tibia/pin to different positions, we attached a rare earth magnet (1 cm height × 5 mm diameter column) to a steel post (M3 × 20 mm flat head machine screw) and controlled its position using a programmable servo motor (SilverMax QCI-X23C-1; Max speed 533,333°/s, Max acceleration 83,333.33°/$s^2$, Position resolution 0.045°; QuickSilver Controls). To move the magnet in a circular trajectory centered at the femur-tibia joint, we placed the motor on a micromanipulator (MP-285, Sutter Instruments) and adjusted its position while visually inspecting the movement of the magnet and the tibia using the tibia tracking camera described below. For each trial, we controlled the speed and the position of the servo motor using QuickControl software (QuickSilver Controls). During all trials, we tracked the tibia position (as described below) to confirm the tibia movement during each trial. Because it was difficult to fully flex the femur-tibia joint without the tibia/pin and the magnet colliding with the abdomen, we only flexed the joint up to ~18°. We set the acceleration of the motor to 72,000°/$s^2$ for all ramp and hold and swing movements. Movements

of the tibia during each trial varied slightly due to several factors, including the length of the magnetic pin and the positioning of the tibia and motor.

## Tracking the femur-tibia joint angle during electrophysiology and imaging experiments

To track the position of the tibia, we illuminated the tibia/pin with an 850 nm IR LED (M850F2, Thor-Labs) and recorded video using an IR sensitive high-speed video camera (Basler Ace A800-510um, Basler AG) with a 1.0× InfiniStix lens (94 mm wd, Infinity). The camera used in the calcium imaging prep was further equipped with a 900 nm short pass filter (Edmund Optics) to filter out the two-photon laser light. In order to synchronize the tibia movement with the recorded cell activity, the camera exposure signal was acquired at 20 kHz. To track the tibia angle, we identified the position of the painted tibia/pin against the contrasting background by thresholding the image. We then approximated the orientation of the leg as the long axis of an ellipse with the same normalized second central moments as the thresholded image (*Haralick and Shapiro, 1992*).

## Vibrating the tibia using a piezoelectric crystal

To vibrate the tibia at high frequencies, we moved the magnet using either a piezoelectric crystal (calcium imaging prep, PA3JEW, Max displacement 1.8 µm; ThorLabs) or a preloaded piezoelectric actuator (patch-clamp electrophysiology prep, P-841.40, Physik Instrumente). To control the movement of the piezo, we generated sine waves of different frequencies in MATLAB (sampling frequency 10 kHz) and sent them to the piezo through a single channel open-loop piezo controller (calcium imaging prep: Thorlabs; electrophysiology prep: Physik). Piezo-induced tibia movements during the calcium imaging prep were calibrated as described by *Mamiya et al., 2018*. Piezo-induced movements during the electrophysiology prep were calibrated using the amplitude measured by the piezo's internal sensor. For each stimulus, we presented 4 s of vibration two to three times with an inter-stimulus interval of 8 s. We averaged the responses within each fly before averaging across flies.

## Spike detection from whole-cell recordings

To detect spikes in current clamp recordings of membrane potential, we applied the following analysis steps to our records of membrane voltage: (1) filter, (2) identify events with large peaks above a threshold, (3) compute a distance from a template for each event, (4) compute the amplitude of the voltage deflection associated with the filtered event, (5) select spikes by thresholding events based both on the distance to the filtered template (<threshold) and on the amplitude of the spike in the voltage record (>threshold). The parameter space for each of these steps was explored in an interactive spike detection interface which can be found at https://github.com/tony-azevedo/spikeDetection. Further details regarding the spike detection algorithm can be found in *Azevedo et al., 2020*.

## Immunohistochemistry and anatomy

For confocal imaging, we crossed flies carrying the Gal4 driver to flies carrying *pJFRC7-20XUAS-IVS-mCD8::GFP* and dissected the VNC out of the thorax in *Drosophila* saline. We first fixed the VNC in a 4% paraformaldehyde PBS solution for 15 min and then rinsed the VNC in PBS three times. We next put the VNC in blocking solution (5% normal goat serum in PBS with 0.2% Triton-X) for 20 min, and then incubated it with a solution of primary antibody (anti-CD8 rat antibody 1:50 concentration; anti-brp mouse for nc82 neuropil staining; 1:50 concentration) in blocking solution for 24 hr at room temperature. At the end of the first incubation, we washed the VNC with PBS with 0.2% Triton-X (PBST) three times, and then incubated the VNC in a solution of secondary antibody (anti-rat-Alexa 488 1:250 concentration; anti-mouse-Alexa 633 1:250 concentration) dissolved in blocking solution for 24 hr at room temperature. Finally, we washed the VNC in PBST three times and then mounted it on a slide with Vectashield (Vector Laboratories). Following electrophysiology recordings, we dissected the VNC and brain and followed the procedure described above, but included streptavidin AlexaFluor conjugate (1:250 goat anti-mouse AlexaFluor conjugate from Invitrogen) during the secondary antibody staining to visualize Neurobiotin-filled neurons. We acquired a z-stack image of the slides on a confocal microscope (Zeiss 510).

Cells were traced in FIJI (*Schindelin et al., 2012*), using the Simple Neurite Tracing plug-in (*Longair et al., 2011*). For *in silico* overlay of the expression patterns of specific Gal4 lines (*Figure 1I–K*), we used confocal stacks of each Gal4 line with neuropil counterstaining (from the Janelia FlyLight database *Jenett et al., 2012*) and used the neuropil staining to align the expression pattern in the VNC using the Computational Morphometry Toolkit (*Jefferis et al., 2007*; http://nitrc. org/projects/cmtk) to a female VNC template (*Bogovic et al., 2019*, Janelia Research Campus, https://www.janelia.org/open-science/jrc-2018-brain-templates).

## Pharmacology

Drugs were bath applied via the saline perfusate. TTX (purchased from Abcam) was prepared as a concentrated stock solution in sodium citrate, picrotoxin was prepared as a concentrated stock solution in aqueous NaCl (140 mM), and methyllycaconitine citrate (MLA, purchased from Sigma-Aldrich) and atropine sulfate (Sigma-Aldrich) were prepared as stock solutions in water. Each drug was further diluted in saline for experiments for a final concentration of 1 μM (TTX and MLA), 20 μM (atropine), or 100 μM (picrotoxin). Drugs were perfused over the exposed VNC for as long as 40 min (MLA and atropine in the case of 13Bα and 10A cell recordings) but more often for 20 min.

## Fly preparation for walking experiments

Fly wings were clipped under cold anesthesia (<4 min) 24 hr before walking experiments. The fly's dorsal thorax was attached to a tungsten wire (0.1 mm diameter) with UV-curing glue (KOA 300, KEMXERT). Tethered flies were food deprived for at least 3 hr prior to being placed in the arena. In the headless preparation, the tethered flies were then decapitated under cold anesthesia and allowed to recover for 5–10 min prior to the experiment. Intact or headless tethered flies were positioned on a hand-milled foam treadmill ball (density: 7.3 mg/mm$^3$, diameter: 9.46 mm) that was suspended on a stream of air (5 l/min) and freely rotated under the fly's movement. The ball and fly were illuminated by three IR lights (M850F2, ThorLabs) to improve motion tracking. In unloaded experiments with the headless prep, we removed the spherical treadmill, leaving the flies suspended in air. For all trials, the temperature in the chamber was maintained between 26 and 28°C with a relative humidity of 58–65%.

## Tethered behavior assay

We coaxed flies to walk on the ball by displaying visual stimuli on a semi-circular green LED display (*Reiser and Dickinson, 2008*). To elicit forward walking, we displayed a single dark bar (width 30°) on a light background, and sinusoidally oscillated the bar at 2.7 Hz across 48.75° about the center of the fly's visual field. During periods between trials, the LED panels displayed a fixed dark stripe (30°) on a bright background in front of the tethered fly. To characterize the role of the motor neurons in behaving tethered flies, we optogenetically activated genetically targeted motor neurons. A green laser (532 nm, CST DPSS laser, Besram Technology, Inc), pulsed at 1200 Hz with a 66% duty cycle, passed through a converging lens and a pinhole (50 μm diameter) with a resulting power of 87 mW/mm$^2$ at the target. It was aimed at the fly's left prothoracic coxa-body wall joint, thus targeting the left T1 neuromere below the cuticle. Experiments using a driver line labeling all motor neurons (OK371-Gal4) indicated that optogenetic stimulation primarily affected neurons innervating the left prothoracic leg (*Azevedo et al., 2020*), though we cannot rule out effects on other VNC neurons.

For intact fly experiments, each trial was 4 s long. We presented walking flies with the visual stimulus, the flies reached a steady running speed at ~1.5 s, and the laser stimulus began at 2 s. The laser stimulus randomly cycled through seven stimulus lengths: 0 ms, 30 ms, 60 ms, 90 ms, 180 ms, 360 ms, and 720 ms. For simplification, we primarily focus on a short stimulus (90 ms), a long stimulus (720 ms), and the control condition (laser omitted, 0 ms). Each fly was presented each laser stimulus six times. For headless fly experiments, we used only the longest laser stimulus length (720 ms) and the control omitted stimulus (0 ms), such that each fly had 24 laser stimulus trials (at 720 ms) and four control trials (no laser), randomly interleaved. Trials were separated by a 25 s period during which video data was written to disk and the LED panels displayed a fixed, stationary stripe.

## Quantification of fly behavior

We used Fictrac (*Moore et al., 2014*) to calculate fly walking trajectories (position, speed, and rotational velocity) from live video of the spherical treadmill's rotation (Point Grey Firefly camera, imaging at 30 Hz). Trajectories were then converted from pixels to mm using the spherical treadmill's diameter of 9.46 mm. Detailed fly movements and kinematics were captured from six simultaneously triggered cameras (Basler acA800-510µm, imaging at 300 Hz) that were distributed around the fly. Digital and analog data signals were collected with a DAQ (PCIe-6321, National Instruments) sampling at 10 kHz and recorded with custom MATLAB scripts. For all experimental trials, we scored the fly's behavior in the 200 ms preceding the optogenetic stimulus as stationary, walking/turning, grooming or other. Flies that took no steps for the duration of the categorization period were classified as stationary. Flies that took at least four coordinated steps over the duration of the 200 ms period were classified as walking/turning. Trials in which the fly switched behaviors, groomed, or did not display clear markers for walking/turning during the categorization period were classified as other/grooming and excluded from analyses. For each headless fly trial, both unloaded and loaded, we also scored the behavioral response to the laser stimulus during the 720 ms period following the onset of the stimulus into categories based on the repertoire of responses.

In headless fly experiments, we manually tracked the position of the left front leg via high-speed video during the optogenetic stimulus period. We then calculated the leg joint angles (coxa-femur, femur-tibia, and tibia-tarsus) from the position measurements. For activation experiments in 13Bα headless flies, we calculated the average change in the leg joint angles (coxa-femur, femur-tibia, and tibia-tarus) over time across flies for the control (0 ms laser) and activation stimulus (720 ms laser). We calculated the change in joint angle as the difference in the average joint angle for a 200 ms period before the laser turned and the last 200 ms period of the laser activation. For activation experiments in 9Aα, we subtracted the joint angle during the frame immediately preceding laser onset from the joint angle during the frame immediately following laser offset, excluding any flies that were not stationary in the 200 ms preceding the optogenetic stimulus. We then compared this change in joint angle during trials with a 720 ms laser stimulus with trials with a 0 ms laser stimulus using bootstrap simulations with 100,000 random draws to compare changes in walking speed (*Saravanan and Berman, 2020*). We calculated the change in joint angle over time for all trials binned by initial joint position (e.g. *Figure 3E and F*) to determine if the initial position of the joint affected the response to activation. Initial joint angles were determined from the camera frame before the laser stimulus started. Comparisons across these different groups were accomplished using a one-way ANOVA with Tukey–-Kramer corrections.

To calculate the maximum joint flexion that occurred due to 13Bα activation, we used all trials in which the fly was subject to the 720 ms laser activation period and excluded flies that were not stationary during the 200 ms preceding the stimulus. Maximum joint flexion is the change in initial joint angle (×) minus the minimum joint angle achieved during the first 500 ms of the laser activation period.

For walking fly experiments (e.g. *Figures 8* and S8), we calculated the average forward velocity over time in the walking trials for each stimulus length, for each fly. We noticed a response in the control flies (split-half [SH]-Gal4) to the laser light that was correlated with the laser turning off; thus, we compared changes in speed between the control line (SH-Gal4) and each of the central neuron lines. We first calculated the change in running speed within a genotype as the average difference in speed between the 200 ms period preceding the laser stimulus and the 200 ms period occurring 200 ms after the laser onset (black brackets in *Figure 8C*). We calculated the difference between the change in running speed for each stimulus condition in the control line and the interneuron line (*Figure 8B*). We then used bootstrap simulations with 100,000 random draws to compare changes in walking speed for a given central neuron line to the control. To quantify the pausing/stopping behavior observed during activation of the 10Bα neurons, we generated a cumulative probability distribution for the first instance of freezing (speed dropping below 0.3 cm/s) within the 2-s period following the laser stimulus start for each walking trial (*Figures 8D* and S8C).

## Statistical analyses

For electrophysiology and calcium imaging results, no statistical tests were performed *a priori* to decide upon sample sizes, but sample sizes were consistent with conventions in the field. Unless

otherwise noted, we used the nonparametric Wilcoxon matched-pairs signed-rank test. We compared cell activity before and after drug application using a Wilcoxon matched-pairs signed-rank test.

For headless fly behavior experiments, we compared changes in joint angle during trials with a 720 ms laser stimulus with trials with a 0 ms laser stimulus using bootstrap simulations with 100,000 random draws to compare changes in walking speed (*Saravanan and Berman, 2020*). We calculated the change in joint angle over time for all trials binned by initial joint position (e.g. *Figure 3E and F*) to determine if the initial position of the joint affected the response to activation. Initial joint angles were determined from the camera frame before the laser stimulus started. Comparisons across these different groups was accomplished using a one-way ANOVA with Tukey–Kramer corrections. For walking fly experiments, we also used bootstrap simulations with 100,000 random draws to compare changes in walking speed for a given central neuron line to the control and then compared these different groups using a one-way ANOVA with Tukey–Kramer corrections. We used the Benjamini–Hochberg procedure to calculate the false-discovery-rate.

## Table of genotypes

| | |
|---|---|
| *Figure 1A* | W[1118]; P{JFRC7-20XUAS-IVS-mCD8::GFP} attp40/+; iav-Gal4/+ |
| *Figure 1C* | Left: w[1118]; P{JFRC7-20XUAS-IVS-mCD8::GFP} attp40/+; P{GMR73D10-GAL4}attP2/+ Right: w[1118]/+; P{20xUAS-IVS-GCaMP6f} attP40 /+; P{GMR73D10-GAL4}attP2/P{w[+mc]=UAS-tdTomato} |
| *Figure 1D* | Left: w[1118]; P{JFRC7-20XUAS-IVS-mCD8::GFP} attp40/+; P{GMR21D12-GAL4}attP2/+ Right: w[1118]/+; P{20xUAS-IVS-GCaMP6f} attP40; P{GMR21D12-GAL4}attP2/P{w[+mc]=UAS-tdTomato} |
| *Figure 1E* | Left: w[1118]; P{JFRC7-20XUAS-IVS-mCD8::GFP} attp40/+; P{GMR64C04-GAL4}attP2/+ Right: w[1118]/+; P{20xUAS-IVS-GCaMP6f} attP40 /+; P{GMR64C04-GAL4}attP2/P{w[+mc]=UAS-tdTomato} |
| *Figure 1F* | Left: w[1118]; P{JFRC7-20XUAS-IVS-mCD8::GFP} attp40/P{28A12-p65.AD} attp40; P{VT000606-GAL4.DBD}attP2/+ Right: w[1118]; P{20xUAS-IVS-GCaMP6f} attP40/P{28A12-p65.AD} attp40; P{VT000606-GAL4.DBD}attP2/P{w[+mc]=UAS-tdTomato} |
| *Figure 1G* | Left: w[1118]; P{JFRC7-20XUAS-IVS-mCD8::GFP} attp40/P{52E12-p65.AD} attp40; P{VT044946-GAL4.DBD}attP2/+ Right: w[1118]; P{20xUAS-IVS-GCaMP6f} attP40/P{52E12-p65.AD} attp40; P{VT044946 -GAL4.DBD}attP2/P{w[+mc]=UAS-tdTomato} |
| *Figure 1H* | Left: w[1118]; P{JFRC7-20XUAS-IVS-mCD8::GFP} attp40/P{VT043132-p65.AD} attp40; P{VT045623-GAL4.DBD}attP2/+ Right: w[1118]; P{20xUAS-IVS-GCaMP6f} attP40/P{VT043132 -p65.AD} attp40; P{VT045623-GAL4.DBD}attP2/P{w[+mc]=UAS-tdTomato} |
| *Figure 2* and *Figure 2— figure supplement 1* | Recordings obtained from two different lines: w[1118]; P{JFRC7-20XUAS-IVS-mCD8::GFP} attp40/P{28A12-p65.AD} attp40; P{VT000606-GAL4.DBD}attP2/+ w[1118]; P{JFRC7-20XUAS-IVS-mCD8::GFP} attp40/ +; P{25G05-GAL4}attP2/+ |
| *Figure 3* and *Figure 3— figure supplement 1* | w[1118]; P{28A12-p65.AD} attp40/+; P{VT000606-GAL4.DBD}attP2/P{y[+t7.7] w[+mC]=20 XUAS-IVS-CsChrimson.mVenus}attP2 |
| *Figures 4* and *5*, *Figure 4— figure supplement 1*, and *Figure 5— figure supplement 1* | w[1118]; P{JFRC7-20XUAS-IVS-mCD8::GFP} attp40/P{52E12-p65.AD} attp40; P{VT044946-GAL4.DBD}attP2/+ |

*Continued on next page*

| | |
|---|---|
| *Figure 6* | w[1118]; P{52E12-p65.AD} attp40/+; P{VT044946-GAL4.DBD}attP2/P{y[+t7.7] w[+mC]=20 XUAS-IVS-CsChrimson.mVenus}attP2 |
| *Figure 7* and *Figure 7— figure supplement 1* | w[1118]; P{JFRC7-20XUAS-IVS-mCD8::GFP} attp40/P{VT043132-p65.AD} attp40; P{VT045623-GAL4.DBD}attP2/+ |
| *Figure 8* | Control (SH-Gal4): w[1118]; P{10A07-p65.AD} attp40/+; P{y[+t7.7] w[+mC]=20 XUAS-IVS-CsChrimson.mVenus}attP2/+ 10Bα-Gal4: w[1118]; P{VT043132-p65.AD} attp40/+; P{VT045623-GAL4.DBD}attP2/P{y[+t7.7] w[+mC]=20 XUAS-IVS-CsChrimson.mVenus}attP2 13Bα-Gal4: w[1118]; P{28A12-p65.AD} attp40/+; P{VT000606-GAL4.DBD}attP2/P{y[+t7.7] w[+mC]=20 XUAS-IVS-CsChrimson.mVenus}attP2 9Aα-Gal4: w[1118]; P{52E12-p65.AD} attp40/+; P{VT044946-GAL4.DBD}attP2/P{y[+t7.7] w[+mC]=20 XUAS-IVS-CsChrimson.mVenus}attP2 |
| *Figure 1— figure supplement 1A* | w[1118]; P{JFRC7-20XUAS-IVS-mCD8::GFP} attp40/P{28A12-p65.AD} attp40; P{VT000606-GAL4.DBD}attP2/+ |
| *Figure 1— figure supplement 1B* | w[1118]; P{JFRC7-20XUAS-IVS-mCD8::GFP} attp40/P{28A12-p65.AD} attp40; P{VT000606-GAL4.DBD}attP2/+ |
| *Figure 1— figure supplement 1C* | w[1118]; P{JFRC7-20XUAS-IVS-mCD8::GFP} attp40/P{VT043132-p65.AD} attp40; P{VT045623-GAL4.DBD}attP2/+ |
| *Figure 1—figure supplement 1D–F* | w[1118]; P{20xUAS-IVS-GCaMP6f} attP40/P{28A12-p65.AD} attp40; P{VT000606-GAL4.DBD}attP2/P{w[+mc]=UAS-tdTomato} |
| *Figure 1—figure supplement 1G–I* | w[1118]; P{20xUAS-IVS-GCaMP6f} attP40/P{52E12-p65.AD} attp40; P{VT044946 -GAL4.DBD}attP2/P{w[+mc]=UAS-tdTomato} |
| *Figure 1—figure supplement 1J–L* | w[1118]; P{20xUAS-IVS-GCaMP6f} attP40/P{VT043132 -p65.AD} attp40; P{VT045623-GAL4.DBD}attP2/P{w[+mc]=UAS-tdTomato} |
| *Figure 6—figure supplement 1* | w[1118]; P{VT014013-p65.AD} attp40/+; P{30A10-GAL4.DBD}attP2/P{y[+t7.7] w[+mC]=20 XUAS-IVS-CsChrimson.mVenus}attP2 |
| *Figure 8—figure supplement 1A* | Control (SH-Gal4): w[1118]; P{10A07-p65.AD} attp40/+; P{y[+t7.7] w[+mC]=20 XUAS-IVS-CsChrimson.mVenus}attP2/+ 10Bα-Gal4: w[1118]; P{VT043132-p65.AD} attp40/+; P{VT045623-GAL4.DBD}attP2/P{y[+t7.7] w[+mC]=20 XUAS-IVS-CsChrimson.mVenus}attP2 |
| *Figure 8—figure supplement 1B and C* | 13Bα-Gal4: w[1118]; P{28A12-p65.AD} attp40/+; P{VT000606-GAL4.DBD}attP2/P{y[+t7.7] w[+mC]=20 XUAS-IVS-CsChrimson.mVenus}attP2 9Aα-Gal4: w[1118]; P{52E12-p65.AD} attp40/+; P{VT044946-GAL4.DBD}attP2/P{y[+t7.7] w[+mC]=20 XUAS-IVS-CsChrimson.mVenus}attP2 9Aα-L2-Gal4: w[1118]; P{VT014013-p65.AD} attp40/+; P{30A10-GAL4.DBD}attP2/P{y[+t7.7] w[+mC]=20 XUAS-IVS-CsChrimson.mVenus}attP2 |

## Acknowledgements

We thank Haluk Lacin, Anthony Azevedo, and Akira Mamiya for helpful discussions, members of the Tuthill laboratory for feedback on the manuscript, Peter Detwiler, Fred Rieke, and Rachel Wong for generous sharing of equipment, Shellee Cunnington for preparation of solutions, Akira Mamiya, Eric Martinson, and Bryan Venema for technical assistance, and Michael Dickinson for sharing fly stocks. We used stocks obtained from the Bloomington *Drosophila* Stock Center (NIH P40OD018537). We also acknowledge support from the NIH (S10 OD016240) to the Keck Imaging Center at UW, and the assistance of its manager, Nathaniel Peters. ED and PG were partially funded by post-baccalaureate fellowships from the UW Institute for Neuroengineering (UWIN). This work was supported by the UW Royalty Research Fund, a UW Innovation Award, a Searle Scholar Award, a Klingenstein-

Simons Fellowship, a Pew Biomedical Scholar Award, a Sloan Research Fellowship, and NIH grant R01NS102333 to JCT.

## Additional information

### Funding

| Funder | Grant reference number | Author |
| --- | --- | --- |
| National Institutes of Health | R01NS102333 | Sweta Agrawal<br>Evyn S Dickinson<br>Anne Sustar<br>Pralaksha Gurung<br>John C Tuthill |
| Howard Hughes Medical Institute | | David Shepherd<br>James W Truman |
| Pew Charitable Trusts | Scholar Award | Sweta Agrawal<br>Evyn S Dickinson<br>Anne Sustar<br>Pralaksha Gurung<br>John C Tuthill |
| Searle Scholars Program | Scholar Award | Sweta Agrawal<br>Evyn S Dickinson<br>Anne Sustar<br>Pralaksha Gurung<br>John C Tuthill |
| Alfred P. Sloan Foundation | Scholar Award | Sweta Agrawal<br>Evyn S Dickinson<br>Anne Sustar<br>Pralaksha Gurung<br>John C Tuthill |
| McKnight Endowment Fund for Neuroscience | Scholar Award | Sweta Agrawal<br>Evyn S Dickinson<br>Anne Sustar<br>Pralaksha Gurung<br>John C Tuthill |
| UW Royalty Research Fund | 112375 | Sweta Agrawal<br>Pralaksha Gurung<br>John C Tuthill |
| Klingenstein Simon Fellowship Award | | Sweta Agrawal<br>Evyn S Dickinson<br>Anne Sustar<br>Pralaksha Gurung<br>John C Tuthill |

The funders had no role in study design, data collection and interpretation, or the decision to submit the work for publication.

### Author contributions

Sweta Agrawal, Conceptualization, Data curation, Formal analysis, Investigation, Visualization, Methodology, Writing - original draft, Writing - review and editing; Evyn S Dickinson, Data curation, Software, Formal analysis, Investigation, Visualization, Writing - review and editing; Anne Sustar, Investigation, Visualization; Pralaksha Gurung, Investigation; David Shepherd, James W Truman, Resources; John C Tuthill, Conceptualization, Supervision, Funding acquisition, Validation, Investigation, Writing - review and editing

### Author ORCIDs

Sweta Agrawal (iD) https://orcid.org/0000-0003-0547-4099
Evyn S Dickinson (iD) http://orcid.org/0000-0001-7518-9512
Anne Sustar (iD) https://orcid.org/0000-0002-9821-7456
Pralaksha Gurung (iD) https://orcid.org/0000-0002-1189-1393

David Shepherd (iD) http://orcid.org/0000-0002-6961-7880
James W Truman (iD) http://orcid.org/0000-0002-9209-5435
John C Tuthill (iD) https://orcid.org/0000-0002-5689-5806

**Decision letter and Author response**
Decision letter https://doi.org/10.7554/eLife.60299.sa1
Author response https://doi.org/10.7554/eLife.60299.sa2

## Additional files

### Supplementary files

- Transparent reporting form

### Data availability

Data made freely available on Dryad (https://doi.org/10.5061/dryad.k3j9kd55t).

The following dataset was generated:

| Author(s) | Year | Dataset title | Dataset URL | Database and Identifier |
|---|---|---|---|---|
| Agrawal S, Dickinson ES, Sustar A, Gurung P, Shepherd D, Truman JW, Tuthill JC | 2020 | Central processing of leg proprioception in Drosophila: physiology and behavior data | http://dx.doi.org/10.5061/dryad.k3j9kd55t | Dryad Digital Repository, 10.5061/dryad.k3j9kd55t |

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
