## [Decision Letter]

**Acceptance summary:**

How the central nervous system coordinates motor control in response to proprioceptive input, an important feedback signal, is a big question in neuroscience. *Drosophila* presents a powerful model to tackle this question. Agrawal et al. carefully characterized physiological properties and behavioral contributions of 3 types of newly identified putative 2nd order proprioceptive cells. The results show that central circuits integrate information across proprioceptor subtypes to construct complex sensorimotor representations that are critical for reflexive control of limb posture. These findings are very significant and useful for future study in the fly model and have general implications for motor control.

**Decision letter after peer review:**

Thank you for submitting your article "Central processing of leg proprioception in *Drosophila*" for consideration by *eLife*. Your article has been reviewed by three peer reviewers, and the evaluation has been overseen by Ronald Calabrese as the Senior and Reviewing Editor. The following individuals involved in review of your submission have agreed to reveal their identity: Eve Marder (Reviewer #1); Terufumi Fujiwara (Reviewer #3).

The reviewers have discussed the reviews with one another and the Reviewing Editor has drafted this decision to help you prepare a revised submission.

Summary:

How the central nervous system coordinates motor control in response to proprioceptive input, an important feedback signal, is a big question in neuroscience. *Drosophila* presents a powerful model to tackle this question, yet how proprioception is processed in the fly's central nervous system is not well known. Agrawal et al. carefully characterized physiological properties and behavioral contributions of 3 types of newly identified putative 2nd order proprioceptive cells. These findings are very significant and useful for future study in the fly model. The analysis is both careful and promising but would benefit from a more explicit discussion of how that sensory information is actually transformed or combined on its way through the central circuits. Then the paper would be a more edifying and generally relevant read.

Revisions:

1) The authors should undertake a significant rewrite to more clearly explain the relevance of the finding for the general field of motor control. This can be accomplished by discussing what has been learned about the sensory transformation in this system and by incorporating some of what is known from the study of large insects and vertebrates and thus putting the findings in context.

2) There were specific concerns about the limitations and interpretability of pharmacological (and other) experiments particularly on the 13B neurons. These can be found in the expert reviews and should be thoroughly addressed.

3) All major concerns of the reviewers should be addressed.

Reviewer #1:

Overview: This is an extensive description of the identification and roles of several classes of proprioceptor neurons in *Drosophila*. Of necessity, a fair bit of the manuscript deals with fly idiosyncratic methods and approaches, and I found it a bit hard work because it wasn't always as clear as it could be between what has been achieved in this work, and what had been done or surmised in the older work on large insects that benefited by classical neuroanatomical, electrophysiological, and behavioral work. Indeed, throughout, even into the Discussion, it was hard to find the "big picture" results or conclusions from this work. Of course, I understand that eventual understanding of how the fly uses these neurons and their particular properties will await better understanding of the rest of the circuit. Nonetheless, I am still struggling to walk away from this manuscript with something a bit more conceptual. I realize it is probably embedded in manuscript, but if so, it is well embedded. Even the first paragraph of the Discussion jumps right into the details.

1) It is fairly common that the concentration of antagonists needed to block synaptically released transmitter can be much higher than the dose needed to apply the same transmitter applied exogenously. This can occur if the synaptically released substance hits the receptors at very high (and unknown) local concentrations. So in general, lack of efficacy of antagonists at a synapse is not a reliable way of arguing transmitter identity at that synapse. This calls into question the strong interpretation argued in paragraph four of subsection “13Bα neurons linearly encode tibia position via tonic changes in membrane potential”.

2) In Figure 5D,E. Is that an n=3 in control and drug? Is that a large enough sample size? In general, throughout the manuscript, please check that appropriate stats are always included?

Reviewer #2:

Sensory information about limb position is detected by proprioceptors, processed by central circuits, and fed back onto motor neurons to guide postural adjustment and movement. Previous work from the Tuthill lab identified proprioceptors (in the femoral chordotonal organ at the femur-tibial joint) and determined what limb movements, joint angles, and vibration they encode. Here Agrawal et al. locate central neurons from three different developmental lineages in the ventral nerve cord that are in anatomical positions to receive these proprioceptive inputs and determine that they respond to aspects of leg movement.

Some of the central neurons respond to multiple different kinds of proprioceptive inputs, suggesting that different sensory neurons may converge onto these neurons, while others are more specific. Figure 8E shows a nice summary of these proposed convergence pathways.

Comparing the response properties of the central neurons to those of the previously-characterized sensory neurons using GCaMP imaging and whole-cell patch electrophysiology suggests some minimal signal processing occurs.

Optogenetic activation of the central neurons causes leg movements. Induced behaviors tend to counter the maximal sensory response (e.g. activating a neuron that encodes joint extension induces flexion), suggesting that these central neurons form part of feedback circuits that correct limb position.

This is elegant and careful work that furthers our understanding of how neural circuits process proprioceptive sensory information and use it for motor control.

Reviewer #3:

How the central nervous system coordinates motor control is a big question in neuroscience. Though *Drosophila* is an exciting model to challenge this question, how proprioception, an important feedback signal for motor control is processed in the central nervous system is not well known. Agrawal et al., beautifully characterized physiological properties and behavioral contributions of 3 types of newly identified putative 2nd order proprioceptive cells. I believe these findings are very significant and useful for future study in the field and therefore.

1) "In contrast to the sensory neurons, nothing is known about how proprioceptive signals are combined or transformed by downstream circuits in the *Drosophila* central nervous system" This might be true for "the leg system" of "the adult" *Drosophila*.

2) Paragraph four of subsection “13Bα neurons linearly encode tibia position via tonic changes in membrane potential”. In my understanding, FeCO is supposed to have a baseline firing, and when the tibia is flexed, 13Ba cells are hyperpolarized because FeCO decreases the firing rate. Then is it true that after TTX application, the baseline membrane potential of 13Ba cells drops due to lack of constant baseline input from FeCO? At least, it appears the case in the example trace of Figure 2—figure supplement 1A. Clarifying this might help answering the following question 3).

3) There are also other inhibitory receptors such as GABAB receptors, so it might still be the case that hyperpolarization during tibia flexion is an inhibitory mechanism. Then, remaining hyperpolarization under MLA can still be explained by an inhibitory mechanism (which is blocked by TTX but not by MLA) instead of gap junction. It would be nice to have additional logic supporting the claim here, if possible.

4) Figure 3. Since laser OFF conditions are used as the control here, to claim that the effect is not simply due to the laser stimulation itself, it is better to briefly mention that the difference between laser ON and OFF conditions is not evident for some cases in different genotype (e.g. Figure 6).

5) Figure 3E. It is interesting that the magnitude of activation effect is independent of the initial angle. When the initial angle is close to 180 degrees, 13Ba cells are supposed to be already quite depolarized. Do you assume that the laser activation further depolarizes the cells to still substantially affect behavior in such condition? Or does the probability of the joint flexion decrease as the initial angle increases?

6) Figure 5D, E. It would be nice to have some speculation why both MLA and picrotoxin abolish excitatory and inhibitory components.

7) Paragraph four of subsection “10Bα neurons drive pausing behavior in walking flies” and paragraph three of subsection “Behavioral function of central proprioceptive neurons”. Do these sentences mean that the source of vibration detected by club FeCO can be both self- and externally- generated? An alternative possibility of the differential behavioral consequence of the activation of 9Aa and 10Ba cells might be due to the difference in their neurotransmitters instead of their encoding. I am not sure if the result of 9Aa activation obviously supports that they encode self-generated vibration.

[Editors' note: further revisions were suggested prior to acceptance, as described below.]

Thank you for resubmitting your article "Central processing of leg proprioception in *Drosophila*" for consideration by *eLife*. Your revised article has been reviewed by two peer reviewers, and the evaluation has been overseen by Ronald Calabrese as the Senior and Reviewing Editor. The reviewers have opted to remain anonymous.

The reviewers have discussed the reviews with one another and the Reviewing Editor has drafted this decision to help you prepare a revised submission.

We would like to draw your attention to changes in our revision policy that we have made in response to COVID-19 (https://elifesciences.org/articles/57162). Specifically, when editors judge that a submitted work as a whole belongs in *eLife* but that some conclusions require further revision and a modest amount of additional data, as they do with your paper.

Summary:

How the central nervous system coordinates motor control in response to proprioceptive input, an important feedback signal, is a big question in neuroscience. *Drosophila* presents a powerful model to tackle this question, yet how proprioception is processed in the fly's central nervous system is not well known. Agrawal et al. carefully characterized physiological properties and behavioral contributions of 3 types of newly identified putative 2nd order proprioceptive cells. These findings are very significant and useful for future study in the fly model. The analysis is both careful and promising but would benefit from a more explicit discussion of how that sensory information is actually transformed or combined on its way through the central circuits. Then the paper would be a more edifying and generally relevant read.

Please focus on supplying additional numbers of animals for the experiments mentioned by reviewer #1 and revise accordingly.

Reviewer #1:

There remain issues which need resolution before publication.

1) The manuscript provides no citations for the block of glutamate Cl channels by picrotoxin in flies (or in any system). There have been data published in crustaceans that could/should be cited, but has it been also done in flies?

2) The authors' response to the queries about the stats in Figure 5D-E is inadequate. If they had success doing this experiment to completion in 6/15 cells, they could have and should increase the n's by just suffering and making a few more recordings….unless there is a really compelling reason why not, which I can't imagine. I am really unmoved by language saying that the authors "believe" something is meaningful, despite such small sample sizes….just get a more robust data set and replace "belief" with "evidence".

3) I am disturbed on the authors' statement about switching to a two-sample t-test on an n=2. Whether or not you can run a statistic on the computer doesn't mean the result is statistically or biologically sensible. We should not be doing or publishing stats on n=2. I don't believe that a two-sample t-test was intended for this situation. The problem remains the same, the sample size is too small.

4) Please do not say that membrane potential "decreases" when you mean "hyperpolarizes"….Decrease is ambiguous. Use either depolarizes or hyperpolarizes in all cases in the manuscript.

5) On rereading, I was wondering whether the authors might wish to think about the crustacean SROs and their role in motor control…as well as the vertebrate spinal cord. This might have been relevant in the Introduction. Not necessary, but worth thinking about.

6) Are there citations for the findings of the transmitters used in 3 classes of neurons (and do they have co-transmitters in them as well?). What is the evidence for those transmitter assignments?

7) I am a confused by Figure 2 F. If I look at the blue traces, it looks like the recovery trace of the up ramp (left) looks similar to the down ramp (right) and likewise in the opposite direction. Then I noticed that the blue traces don't look representative of the gray individual traces. In this figure, what are the blue traces? Are they a selected "best" example, or a mean? Shouldn't the authors tell us f the hysteresis plotted in Figure 2H is the same if you start with a steady-state hold level, or if you compare the up and down traces in the ramps? The legend is uninformative, and the text doesn't exactly match with what the figure shows?

8) A fair bit of the Discussion deals with issues that are only tangential to the actual data in the paper.

9) Subsection “Central integration of proprioceptive sensory information”. Of course the heterogeneity could arise from differences in the synaptic inputs. It could also just be a feature of the neurons themselves.

---

## [Author Response]

Reviewer #1:Overview: This is an extensive description of the identification and roles of several classes of proprioceptor neurons in *Drosophila*. Of necessity, a fair bit of the manuscript deals with fly idiosyncratic methods and approaches, and I found it a bit hard work because it wasn't always as clear as it could be between what has been achieved in this work, and what had been done or surmised in the older work on large insects that benefited by classical neuroanatomical, electrophysiological, and behavioral work. Indeed, throughout, even into the Discussion, it was hard to find the "big picture" results or conclusions from this work. Of course, I understand that eventual understanding of how the fly uses these neurons and their particular properties will await better understanding of the rest of the circuit. Nonetheless, I am still struggling to walk away from this manuscript with something a bit more conceptual. I realize it is probably embedded in manuscript, but if so, it is well embedded…Even the first paragraph of the Discussion jumps right into the details…

The reviewer’s confusion is understandable. We have added a new section to the Discussion contextualizing how our findings relate to the prior work in large insects. Additionally, we have adjusted the Discussion so that we discuss these “big picture” results before diving into other interesting details.

1) It is fairly common that the concentration of antagonists needed to block synaptically released transmitter can be much higher than the dose needed to apply the same transmitter applied exogenously. This can occur if the synaptically released substance hits the receptors at very high (and unknown) local concentrations. So in general, lack of efficacy of antagonists at a synapse is not a reliable way of arguing transmitter identity at that synapse. This calls into question the strong interpretation argued in paragraph four of subsection “13Bα neurons linearly encode tibia position via tonic changes in membrane potential”.

We have updated some of the language to address this concern:

“Surprisingly, both MLA and atropine had only subtle effects on 13Bα encoding, and never completely abolished 13Bα activity (Figure 2—figure supplement 1B-C). This result suggests that either 13Bα neurons are coupled to claw sensory neurons via gap junctions, or that MLA and atropine only partially block cholinergic synaptic input from claw neurons.”

2) In Figure 5D,E. Is that an n=3 in control and drug? Is that a large enough sample size? In general, throughout the manuscript, please check that appropriate stats are always included?

Yes, in Figure 5D-E, our sample size is 3 cells for each the control and drug. Unfortunately, our recordings rarely lasted long enough to map the entire suite of proprioceptive responses, bath apply a drug, and then remap proprioceptive responses post-drug application. For example, of the 15 9Aα cells we recorded from, we were only able to complete this entire sequence for fewer than half of the recordings (6 cells). Of those 6 cells, we applied MLA to half of them and picrotoxin to the other half.

Nevertheless, we believe that this sample size, while low, is still meaningful. First, MLA application almost entirely abolished the vibration response of 9Aα neurons. We observed a similar disruption for both a 9Aα neuron that was inhibited by low frequency vibrations (of a type similar to 5A), and two 9Aα neurons that were not inhibited by low frequency vibrations (of a type similar to 5B). As such, we believe that this sample is representative of the larger 9Aα population. Additionally, as the reviewer noted, such pharmacology experiments can be problematic because we are applying the compound exogenously to the entire fly. There are possible issues about efficacy of the antagonists at the synapse in question, and additionally this manipulation likely affects other neurons in the circuit, meaning that any changes in activity that we observed could be due to either a direct effect of that antagonist on the recorded cell’s synapse, or an indirect effect on an upstream neuron in the circuit.

Regarding the statistics, for all behavior experiments, we made all comparisons, such as the change in joint angle or change in walking speed before and after optogenetic activation by using bootstrap simulations with 100,000 random draws (see Materials and methods for more details). Such a method makes the fewest number of assumptions about the structure of the underlying data (such as normality). Comparisons across different groups were accomplished using a 1-way ANOVA with Tukey-Kramer corrections for multiple comparisons.

We compared cell activity before and after drug application in most cases using a Wilcoxon matched-pairs signed-rank test, except for one case in which the number of cells fell below three (Figure 4—figure supplement 1D-E) in which case we switched to a two sample t-test. We used a matched-pair test since the same cell’s activity from before and after drug application were being compared. The Wilcoxon matched-pairs signed-rank test is a nonparametric test that does not require the underlying data to be normal. However, because it is not robust to smaller sample sizes, we switched to the two sample t-test when the number of cells fell below 3.

Reviewer #3:How the central nervous system coordinates motor control is a big question in neuroscience. Though *Drosophila* is an exciting model to challenge this question, how proprioception, an important feedback signal for motor control is processed in the central nervous system is not well known. Agrawal et al., beautifully characterized physiological properties and behavioral contributions of 3 types of newly identified putative 2nd order proprioceptive cells. I believe these findings are very significant and useful for future study in the field and therefore.1) "In contrast to the sensory neurons, nothing is known about how proprioceptive signals are combined or transformed by downstream circuits in the *Drosophila* central nervous system" This might be true for "the leg system" of "the adult" *Drosophila*.

We have updated the text to be more specific:

“In contrast to the sensory neurons, nothing is known about how leg proprioceptive signals are combined or transformed by downstream circuits in the adult *Drosophila* central nervous system.”

2) Paragraph four of subsection “13Bα neurons linearly encode tibia position via tonic changes in membrane potential”. In my understanding, FeCO is supposed to have a baseline firing, and when the tibia is flexed, 13Ba cells are hyperpolarized because FeCO decreases the firing rate. Then is it true that after TTX application, the baseline membrane potential of 13Ba cells drops due to lack of constant baseline input from FeCO? At least, it appears the case in the example trace of Figure 2—figure supplement 1A. Clarifying this might help answering the following question 3).

Yes, as the reviewer noticed, the membrane potential of 13Bα cells does drop upon TTX application. This is likely because 13Bα neurons receive tonic excitatory input from spiking FeCO neurons that increases or decreases as a function of the femur-tibia joint angle. We have clarified 13Bα connectivity in the relevant paragraph (see response to reviewer #2, major comment #2).

3) There are also other inhibitory receptors such as GABAB receptors, so it might still be the case that hyperpolarization during tibia flexion is an inhibitory mechanism. Then, remaining hyperpolarization under MLA can still be explained by an inhibitory mechanism (which is blocked by TTX but not by MLA) instead of gap junction. It would be nice to have additional logic supporting the claim here, if possible.

As the reviewer has noted, there are other inhibitory receptors that our manipulation does not affect. We have added clarifying language to note this lack, and why we still believe that this manipulation strongly argues against inhibitory input mediating 13Bα activity. Additionally, we have also added in some additional data from current injection experiments that further bolster our conclusion:

“When the femur-tibia joint moves from an extended to a flexed position, the membrane potential of 13Bα neurons decreases to a new steady state. This hyperpolarization could be due to an increase in inhibitory input, a decrease in excitatory input, or a combination of both. Application of picrotoxin, an antagonist of the inhibitory neurotransmitter receptors, GABAA and GluCl, had no effect on 13Bα activity (Figure 2—figure supplement 1D), suggesting that 13Bα neurons do not receive inhibitory input via GABAA or GluCl receptors. Inhibition of 13Bα neurons may instead occur through GABAB receptors that are not blocked by picrotoxin, though GABAB-mediated conductances are slower (Wilson and Laurent, 2005) and therefore unlikely to be involved in encoding rapid joint-angle changes. Further evidence that the hyperpolarization of 13Bα neurons is not mediated by an inhibitory input comes from experiments measuring responses to tibia movement after injecting current to shift the cell’s resting membrane potential (Figure 2—figure supplement 1E-F). Depolarizing the cell by injecting positive current through the patch pipette shifts the membrane potential toward the equilibrium potential for excitation, reducing the driving force for excitatory synaptic input while increasing the driving force for inhibitory synaptic input. Hyperpolarizing the cell by injecting negative current has the opposite effect. We found that when 13Bα cells were depolarized, the change in membrane potential during both extension and flexion decreased (Figure 2—figure supplement 1E), suggesting that the proprioceptive responses of 13Bα neurons are mediated by increases and decreases in excitatory input. In summary, 13Bα cells are a relatively homogeneous class of neurons that receive excitatory input from extension-sensitive claw neurons, perhaps via mixed chemical and electrical synapses.”

The remaining hyperpolarization under MLA still argues for the existence of gap junctions somewhere in the circuit, either between 13Bα and FeCO neurons, or some intermediate neuron that mediates FeCO input onto 13Bα neurons. This is because FeCO neurons are known release acetylcholine.

4) Figure 3. Since laser OFF conditions are used as the control here, to claim that the effect is not simply due to the laser stimulation itself, it is better to briefly mention that the difference between laser ON and OFF conditions is not evident for some cases in different genotype (e.g. Figure 6).

Previous experiments using the same behavioral setup have shown that the laser alone (without expression of CsChrimson) is not sufficient to elicit a behavioral effect in headless flies (Azevedo et al., 2020). We have adjusted the text accordingly:

“In both loaded and unloaded flies, activation of 13Bα neurons caused a slow extension of the coxa-femur joint and flexion of the femur-tibia joint; this movement was absent during trials without a laser stimulus (Figure 3C-D, Video 3). Prior experiments using the same behavioral setup demonstrated that a laser stimulus in the absence of CsChrimson does not cause leg movement in headless flies (Azevedo et al., 2020).”

5) Figure 3E. It is interesting that the magnitude of activation effect is independent of the initial angle. When the initial angle is close to 180 degrees, 13Ba cells are supposed to be already quite depolarized. Do you assume that the laser activation further depolarizes the cells to still substantially affect behavior in such condition? Or does the probability of the joint flexion decrease as the initial angle increases?

We have updated Figures 3 and Figure 3—figure supplement 1 to include the probability of joint flexion for the different initial joint angles (Figure 3G-H), as well as the observed joint flexion as a function of initial joint angle for every trial (Figure 3—figure supplement 1). The probability of joint flexion does decrease a small amount as the joint angle is more extended in both loaded and unloaded flies, though in the case of the unloaded flies, of the 5 trials that had initial joint angles less than 60°, none demonstrated flexion in response to 13Bα activation. Thus, there does not appear to be a consistent relationship between initial joint angle and probability of flexion in our data.

We have updated the relevant text:

“For those flies that flexed their femur-tibia joint, the change in joint angle (Figures 3E-F, Figure 3—figure supplement 1C-D) and likelihood of flexion (Figure 3G-H) did not vary with initial joint position.”

6) Figure 5D, E. It would be nice to have some speculation why both MLA and picrotoxin abolish excitatory and inhibitory components.

MLA abolishes excitatory and inhibitory responses because it blocks all nicotinic acetylcholine receptors, meaning it also blocks the majority of input from the cholinergic FeCO sensory neurons. Picrotoxin’s effect on excitatory components of the 9Aα response is likely because application of picrotoxin depolarized 9Aα cells. This depolarization, in turn, is likely to decrease the driving force for excitatory synaptic inputs. We have included an inset in Figure 5E plotting this change in the membrane potential and updated the text accordingly:

“Picrotoxin also decreased responses to high frequency vibration, but this decrease was likely caused by a reduced excitatory conductance after picrotoxin depolarized the resting membrane potential. Thus, in addition to direction-tuned inputs from FeCO hook neurons, 9Aα cells also receive vibration-sensitive inputs from FeCO club neurons.”

7) Paragraph four of subsection “10Bα neurons drive pausing behavior in walking flies” and paragraph three of subsection “Behavioral function of central proprioceptive neurons”. Do these sentences mean that the source of vibration detected by club FeCO can be both self- and externally- generated? An alternative possibility of the differential behavioral consequence of the activation of 9Aa and 10Ba cells might be due to the difference in their neurotransmitters instead of their encoding. I am not sure if the result of 9Aa activation obviously supports that they encode self-generated vibration.

Yes, some vibrations may be externally generated (arising from external sources, such as wind or movement of other animals), while some vibrations could be internally generated (due to the action of musculoskeletal elements within the fly). Unfortunately, we do not know much about the spectral composition of natural femur-tibia joint movements, and so we are unable to determine how often one or the other of these types of vibrations are experienced by the fly, nor which of these could be detected by the FeCO. We believe that 10Bα neurons encode external vibrations because the behavioral effect of activating 10Bα neurons is similar to a startle effect in response to vibrations of the floor. Additionally, activating 10Bα in headless flies did not cause any joint movements in loaded or unloaded flies. In contrast, 9Aα activation does cause joint movements in headless flies but does not cause a change in walking speed.

[Editors' note: further revisions were suggested prior to acceptance, as described below.]

Reviewer #1:There remain issues which need resolution before publication.1) The manuscript provides no citations for the block of glutamate Cl channels by picrotoxin in flies (or in any system). There have been data published in crustaceans that could/should be cited, but has it been also done in flies?

We have added the relevant citations showing this in flies.

2) The authors' response to the queries about the stats in Figure 5D-E is inadequate. If they had success doing this experiment to completion in 6/15 cells, they could have and should increase the n's by just suffering and making a few more recordings….unless there is a really compelling reason why not, which I can't imagine. I am really unmoved by language saying that the authors "believe" something is meaningful, despite such small sample sizes….just get a more robust data set and replace "belief" with "evidence".

Increasing the sample size of these experiments is technically possible, but would delay resubmission of the manuscript by several months (especially given the decreased research productivity caused by the COVID-19 pandemic and necessity to maintain appropriate safety protocols). We have revised the manuscript so that none of the conclusions in the manuscript hinge on these preliminary data. We also clarify the caveats of interpreting pharmacological manipulations.

3) I am disturbed on the authors' statement about switching to a two-sample t-test on an n=2. Whether or not you can run a statistic on the computer doesn't mean the result is statistically or biologically sensible. We should not be doing or publishing stats on n=2. I don't believe that a two-sample t-test was intended for this situation. The problem remains the same, the sample size is too small.

Upon re-examination, we discovered an unfortunate typo in our Discussion of the statistical tests that we used. We mentioned to both the reviewer and in the Materials and methods section that “We compared cell activity before and after drug application in most cases using a Wilcoxon matched-pairs signed-rank test, except for one case in which the number of cells fell below three (Figure 4—figure supplement 1E) in which case we used a two sample t-test.” In actuality, we only used the two sample t-test in two cases where the number of cells was equal to three, not below three (Figures 4—figure supplement 1E, Figure 7—figure supplement 1F). For the few instances where we had only two data points (namely, experiments in which we applied TTX to either 13Bα, 9Aα, or 10Bα cells), we did not perform any statistical tests because, as the reviewer points out, a statistical test on a sample size that low would be meaningless. We apologize for the confusion, and have corrected this typo. In addition, we have removed the statistics for all cases of n=3, so that now any statistics presented in the manuscript has at least n = 4.

4) Please do not say that membrane potential "decreases" when you mean "hyperpolarizes"….Decrease is ambiguous. Use either depolarizes or hyperpolarizes in all cases in the manuscript.

We have made the suggested change.

5) On rereading, I was wondering whether the authors might wish to think about the crustacean SROs and their role in motor control…as well as the vertebrate spinal cord. This might have been relevant in the Introduction. Not necessary, but worth thinking about.

We agree that the crustacean SROs and their role in motor control is very relevant to the manuscript. SROs, which function similarly to muscle spindles and the FeCO, are a classic and beautiful model for studying mechanosensory transduction and motor control. Unfortunately, in the interest of brevity, we chose to focus on muscle spindles, which is likely more familiar to a broad readership.

6) Are there citations for the findings of the transmitters used in 3 classes of neurons (and do they have co-transmitters in them as well?). What is the evidence for those transmitter assignments?

Yes, the transmitter assignments come from Lacin et al., 2019 (cited in the manuscript) who generated a comprehensive neurotransmitter usage map for the entire ventral nerve cord using molecular and genetic tools. In that study, they found that all neurons within a hemilineage use the same neurotransmitter, suggesting that neurotransmitter identity is acquired at the stem cell level. While they did find evidence that the acetylcholine-specific gene ChAT is transcribed in some glutamatergic and GABAergic neurons, they found that those transcripts did not leave the nucleus nor were they translated. As a result, they concluded that there was no evidence of neurons using more than one neurotransmitter. However, they only examined transcripts relating to the three fast-acting neurotransmitters (acetylcholine, GABA, and glutamate) and did not examine expression of neuropeptides or small molecule neurotransmitters (SMNs). Thus, it is still possible that the 3 classes of neurons may express other neurotransmitters, neuropeptides, or SMNs.

7) I am a confused by Figure 2 F. If I look at the blue traces, it looks like the recovery trace of the up ramp (left) looks similar to the down ramp (right) and likewise in the opposite direction. Then I noticed that the blue traces don't look representative of the gray individual traces. In this figure, what are the blue traces? Are they a selected "best" example, or a mean? Shouldn't the authors tell us f the hysteresis plotted in Figure 2H is the same if you start with a steady-state hold level, or if you compare the up and down traces in the ramps? The legend is uninformative, and the text doesn't exactly match with what the figure shows?

Each trace (gray or blue) is the average response of a single cell to three presentations of the same tibia movement. One example trace is highlighted in blue – it is not a mean trace, simply one that we determined was representative of the population of recorded responses. The hysteresis measured is using a steady-state hold level. Figure 2F is essentially the two traces in Figure 2G subtracted from one another. Figure 2G is generated from the steady-state activity, aka the middle second of each 3-second step of the ramp and hold stimulus. If we were to plot the data from the first second of each step, the hysteresis is very similar, though increased in some cases because many cells exhibited an additional small transient depolarization following joint extension. We have clarified this in the figure legend.

8) A fair bit of the Discussion deals with issues that are only tangential to the actual data in the paper.

We have edited the Discussion to limit tangential digressions.

9) Subsection “Central integration of proprioceptive sensory information”. Of course the heterogeneity could arise from differences in the synaptic inputs. It could also just be a feature of the neurons themselves.

We agree that it could be a feature of the neurons themselves. But as we did not measure features like receptor expression on a cell-by-cell basis, we are proposing that the synaptic inputs could be driving this heterogeneity. In addition, a similar heterogeneity has been observed in a closely related system, the neurons downstream of the *Drosophila* Johnston’s Organ, which is a chordotonal organ in the antenna.